# Estimating effectiveness of case-area targeted response interventions against cholera in Haiti

Edwige Michel[1], Jean Gaudart[2], Samuel Beaulieu[3], Gregory Bulit[3], Martine Piarroux[4], Jacques Boncy[5], Patrick Dely[1], Renaud Piarroux[6]*, Stanislas Rebaudet[7,8]*

[1]Ministry of Public Health and Population, Directorate of Epidemiology Laboratory and Research, Port-au-Prince, Haiti; [2]Aix-Marseille Université, APHM, INSERM, IRD, SESSTIM, Hop Timone, BiSTIC, Biostatistics and ICT, Marseille, France; [3]United Nations Children's Fund, Port-au-Prince, Haiti; [4]Centre d'Épidémiologie et de Santé Publique des Armées, Service de Santé des Armées, Marseille, France; [5]Ministry of Public Health and Population, National Laboratory of Public Health, Delmas, Haiti; [6]Sorbonne Université, Sorbonne Université, INSERM, Institut Pierre-Louis d'Epidémiologie et de Santé Publique (IPLESP), AP-HP, Hôpital Pitié-Salpêtrière, Paris, France; [7]APHM, Hôpital Européen, Aix Marseille Université, INSERM, IRD, SESSTIM, IPLESP, Marseille, France; [8]Sorbonne Université, INSERM, Institut Pierre-Louis d'Epidémiologie et de Santé Publique, Paris, France

**Abstract** Case-area targeted interventions (CATIs) against cholera are conducted by rapid response teams, and may include various activities like water, sanitation, hygiene measures. However, their real-world effectiveness has never been established. We conducted a retrospective observational study in 2015–2017 in the Centre department of Haiti. Using cholera cases, stool cultures and CATI records, we identified 238 outbreaks that were responded to. After adjusting for potential confounders, we found that a prompt response could reduce the number of accumulated cases by 76% (95% confidence interval, 59 to 86) and the outbreak duration by 61% (41 to 75) when compared to a delayed response. An intense response could reduce the number of accumulated cases by 59% (11 to 81) and the outbreak duration by 73% (49 to 86) when compared to a weaker response. These results suggest that prompt and repeated CATIs were significantly effective at mitigating and shortening cholera outbreaks in Haiti.

*For correspondence:
renaud.piarroux@aphp.fr (RP);
stanreb@gmail.com (SR)

Competing interests: The authors declare that no competing interests exist.

## Introduction

On October 2017, Global Task Force on Cholera Control (GTFCC) partners committed to reduce cholera deaths by 90% and to eliminate the disease in 20 countries by 2030, through a multi-sectoral approach (*Anon, 2017*). This new global strategy planned to combine long-term disease prevention in cholera hotspots with sustainable WaSH (water sanitation and hygiene) solutions and large-scale use of oral cholera vaccine (OCV), with the short-term strengthening of early detection of outbreaks and immediate and effective response through reactive OCV campaigns and rapid response teams (RRTs) (*Anon, 2017*). RRTs, also referred as mobile teams, have been successfully implemented against polio or Ebola outbreaks (*Global Polio Eradication Initiative, 2017*; *World Health Organization (WHO), 2014*). However, response interventions targeted to neighbours of cholera cases (case-area targeted interventions [CATIs]) using combinations of water, sanitation, and hygiene measures, and/or prophylactic antibiotics have rarely been documented, evaluated or promoted

against cholera in the published literature (*Voelckel, 1971*; *Piarroux and Bompangue, 2011*; *Deepthi et al., 2013*; *Taylor et al., 2015*; *Mwambi et al., 2016*; *Finger et al., 2018*).

In practice, CATIs are supported by the frequent household transmission of *Vibrio cholerae* O1 (*Weil et al., 2009*; *Weil et al., 2014*; *Taylor et al., 2015*; *Domman et al., 2018*), the increased cholera risk among neighbours living within a few dozen meters of cases during the few days following disease onset (*Debes et al., 2016*; *Azman et al., 2018*), and the significant protection of household contacts of cholera patients by promoting hand washing with soap and treatment of water (*George et al., 2016*). A micro-simulation modelling study suggests that early CATIs can be more resource-efficient than mass interventions against cholera (*Finger et al., 2018*). However, CATI effectiveness has never been evaluated in a real-world setting.

Haiti has implemented CATIs as a national coordinated strategy against cholera since July 2013 (*Rebaudet et al., 2019a*). After the disease was accidentally imported in October 2010 (*Piarroux et al., 2011*), the country experienced a massive epidemic, with a total of 820,085 suspected cases and 9792 cholera-related deaths recorded by April 20, 2019 by the Haitian Ministry of Public Health and Population (MSPP) (http://mspp.gouv.ht/newsite/documentation.php, accessed Jul 1, 2019). In 2013, only 68% of Haitian households drank from improved water sources, 26% had access to improved sanitation facilities and 34% had water and soap available for hand washing (*République d'Haïti, Ministère de la Santé Publique (MSPP), 2013*). But little of the $1.5 billion USD designated by the *Plan for the Elimination of Cholera in Haiti 2013–2022* to develop water and sanitation infrastructures has been expended or pledged so far (*Ministry of Public Health and Population, National Directorate for Water Supply and Sanitation, 2013*). Two pilot OCV reactive campaigns vaccinated approximately 100,000 people in 2012 and to date, additional campaigns have targeted about 10% of the Haitian population (*Ivers, 2017*; *Poncelet, 2015*). UNICEF thus backed the MSPP and the Haitian National Directorate for Water and Sanitation (DINEPA) to launch a complementary nationwide coordinated cholera alert-response strategy aiming to interrupt local cholera outbreaks at an early stage (*Rebaudet et al., 2019a*). This program planned to rapidly send multisectoral rapid response teams to every patient household and neighbourhood in order to identify additional cases, to decontaminate patient premises, to educate on risk factors and methods of prevention and management, to distribute soap and oral rehydration salts (ORS), to chlorinate water at the household level or directly at collection points, and to propose prophylactic antibiotics to close contacts of cholera cases.

This response CATI strategy was implemented gradually from mid-2013 and became an essential pillar of the fight when the national cholera elimination plan was updated in mid-2016 (*République d'Haïti, Ministère de la Santé Publique et de la Population, Direction Nationale de l'Eau Potable et de l'Assainissement, 2016*). Implementation of this strategy offers a unique opportunity to evaluate the effectiveness of CATIs against cholera outbreaks. Based on available data, we conducted a retrospective observational study over 3 years in the Centre department of Haiti addressing the outcome of local cholera outbreaks according to the response promptness and intensity. We present here the first effectiveness estimates for rapid and targeted response interventions against cholera.

## Results

### Outbreak and response characteristics

From January 1, 2015, to December 31, 2017, the line-listing of the Centre department reported a total of 10,931 patients with suspected cholera, including 10,428 with a comprehensive location. Details on cholera cases are summarized in *Appendix 1—table 1*. Intravenous (IV) rehydration was mentioned for 2144 of them. These patients originated from 1497 localities and their time distribution exhibited a marked seasonality (*Figure 1A and B*). Concomitantly, 1070 stools sampled in Centre department were cultured for *V. cholerae* O1, of which 509 (48%) were positive (*Figure 1A*), including 360 with a comprehensive location. Additional details on cholera cultures are summarized in *Appendix 1—table 1*.

Defining outbreaks by the occurrence of at least two suspected cholera cases with at least one severely dehydrated case or positive culture, within the same locality, during a three-day time window, and after a refractory period of at least 21 days with no case, we identified 452 cholera

**Table 1.** Baseline characteristics of outbreaks that were responded to, according to the response promptness (time to the first complete case-area targeted intervention).

| | All outbreaks | Outbreaks responded to with ≥ 1 complete CATI | Class of response promptness (time to the first complete CATI) | | | | Comparison between classes of promptness | |
| --- | --- | --- | --- | --- | --- | --- | --- | --- |
| | | | >7 days | 3 to 7 days | 2 days | ≤1 day | Hazard ratio (95% CI)† | p-value† |
| No. of outbreaks | 452 | 238 (53%) | 48 (20%) | 40 (17%) | 43 (18%) | 107 (45%) | | |
| Semester since January 2015 | | | | | | | 1.10e7 (1.64e6 to 7.40e7) | <0.0001* |
| Population density, median (IQR; inhab./km²) | 3.5 (6.5) | 3.6 (11.5) | 4.3 (10.6) | 2.8 (4.6) | 3.7 (8.9) | 3.8 (12.6) | 1.01 (1 to 1.02) | 0.0039* |
| Travel time to the nearest town, median (IQR; minutes) | 26.7 (33.2) | 24.9 (31.8) | 30 (3 4.3) | 27.1 (42.1) | 24.8 (28.4) | 22 (32) | 1 (0.99 to 1) | 0.274 |
| Accumulated incidence between 2010 and 2014, median (IQR; per 1000 inhabitants) | 103.8 (77.5) | 103.8 (77.5) | 103.8 (131.4) | 103.8 (49.1) | 103.8 (56.6) | 103.8 (77.5) | 0.4 (0.09 to 1.83) | 0.237 |
| Coverage of OCV campaigns between 2012 and 2014, median (IQR; %) [mean, SD] | 0% (86) [25%, 40] | 0% (0) [21%, 38] | 0% (86) [30%, 42] | 0% (0) [18%, 36] | 0% (0) [15%, 33] | 0% (0) [21%, 38] | 0.61 (0.38 to 0.98) | 0.0393* |
| Previous cases in the same locality during the study, median (IQR; no. per year) | 4.3 (10.1) | 5.2 (10.5) | 7.2 (10.1) | 5 (11.5) | 6.9 (11.1) | 5 (8.5) | 0.99 (0.97–1.02) | 0.6540 |
| Previous complete CATIs in the same locality during the study, median (IQR; no. per year) | 0.2 (1.9) | 0.9 (2.7) | 0.7 (2.2) | 0.5 (2.3) | 1.2 (2.9) | 1.4 (2.7) | 0.98 (0.91–1.06) | 0.6500 |
| Daily rainfall during outbreak, median (IQR; mm) | 6.6 (13.3) | 7.7 (13.3) | 12 (6) | 6.9 (10.8) | 10 (13.7) | 3.6 (14.4) | 0.99 (0.96 to 1.03) | 0.638 |
| No. of cases during the first 3 days of outbreak, median (IQR) [mean, SD] | 2 (1) [2.5, 1.5] | 2 (1) [2.7, 1.9] | 2 (1) [2.5, 1.0] | 2.5 (1) [3.4, 2.1] | 2 (0.5) [2.8, 2.4] | 2 (1) [2.5, 1.9] | 1.04 (0.93 to 1.16) | 0.488 |
| No. of positive culture during the first 3 days of outbreak, median (IQR) [mean, SD] | 0 (0) [0.2, 0.6] | 0 (0) [0.3, 0.7] | 0 (0) [0.1, 0.3] | 0 (0) [0.2, 0.5] | 0 (1) [0.5, 0.9] | 0 (1) [0.4, 0.7] | 2.03 (1.3 to 3.17) | 0.0018* |

CATI, case-area targeted intervention; IQR, interquartile range; SD, standard deviation.

†Univariate comparisons between classes of response promptness using Cox models for Andersen-Gill counting process (AG-CP), with time to the first complete CATI modelled as a recurrent time-to-event outcome.

*Significant p-value.

outbreaks (*Figure 2*), which mainly occurred during case incidence peaks (*Figure 1C*) and were distributed across 290 localities (*Figure 3*). The median cumulative number of cases per outbreak was 3 (Interquartile range [IQR], 4), and the median duration of outbreaks was 5 days (IQR, 18).

Over the same period, 3,887 CATIs were notified in the Centre department by non-governmental organization (NGO) rapid response teams, including 3,533 CATIs (91%) with a comprehensive location, and 2719 (70%) conducted in tandem with staff of EMIRA (Equipe mobile d'intervention rapide, *Rapid intervention mobile team*, i.e. cholera rapid response team of the MSPP) (*Figure 1D*). Based on CATI activities summarized in *Appendix 1—table 1*, a total of 3,596 CATIs (93%) were

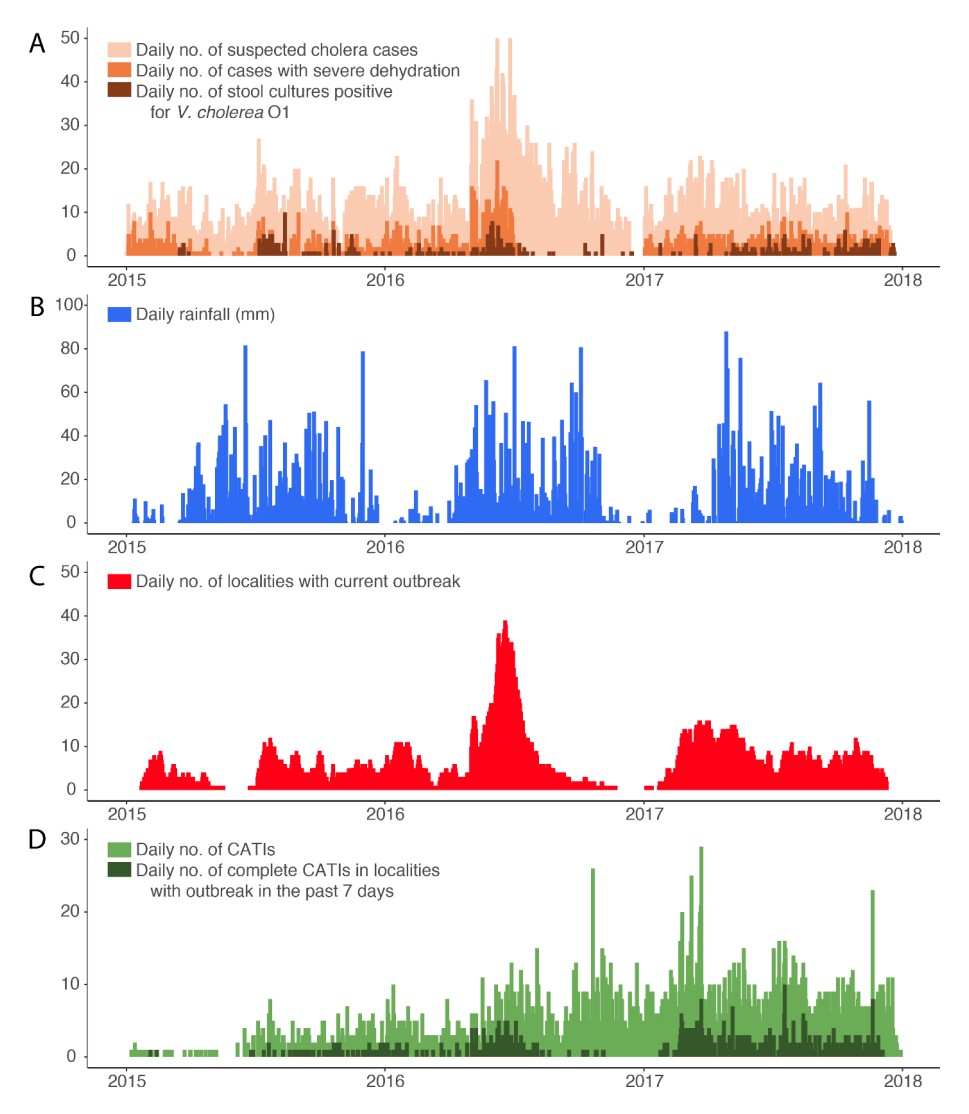

**Figure 1.** Daily evolution of (A) suspected cholera cases, cases with severe dehydration and stool cultures positive for *V. cholerae* O1, (B) accumulated rainfall, (C) localities with a current cholera outbreak, and (D) case-area targeted interventions (CATIs), in the Centre department of Haiti between January 2015 and December 2017. The online version of this article includes the following source data and figure supplement(s) for figure 1:

**Source data 1.** Daily evolution of suspected cholera cases, cases with severe dehydration, stool cultures positive for *V. cholerae* O1, accumulated rainfall, localities with a current cholera outbreak, and case-area targeted interventions (CATIs), between January 2015 and December 2017 in.

**Figure supplement 1.** Daily evolution of suspected cholera cases recorded and stool cultures positive for *V. cholerae* O1 sampled in the Centre department of Haiti between October 2010 and December 2017.

categorized as complete (at least decontamination, education and distribution of chlorine tablets), and 1922 (49%) also included a reported antibiotic prophylaxis. Overall, 633 complete CATIs (18%) were conducted in localities experiencing an identified cholera outbreak (*Figure 1D*).

## Analysis of confounders

Baseline characteristics of outbreaks and comparisons between the four classes of response promptness are presented in *Table 1*. The time to the first complete CATI (response promptness) significantly improved during the six semesters of the study, was significantly higher in more densely populated localities, and was lower in localities targeted by a previous OCV campaign. Outbreaks

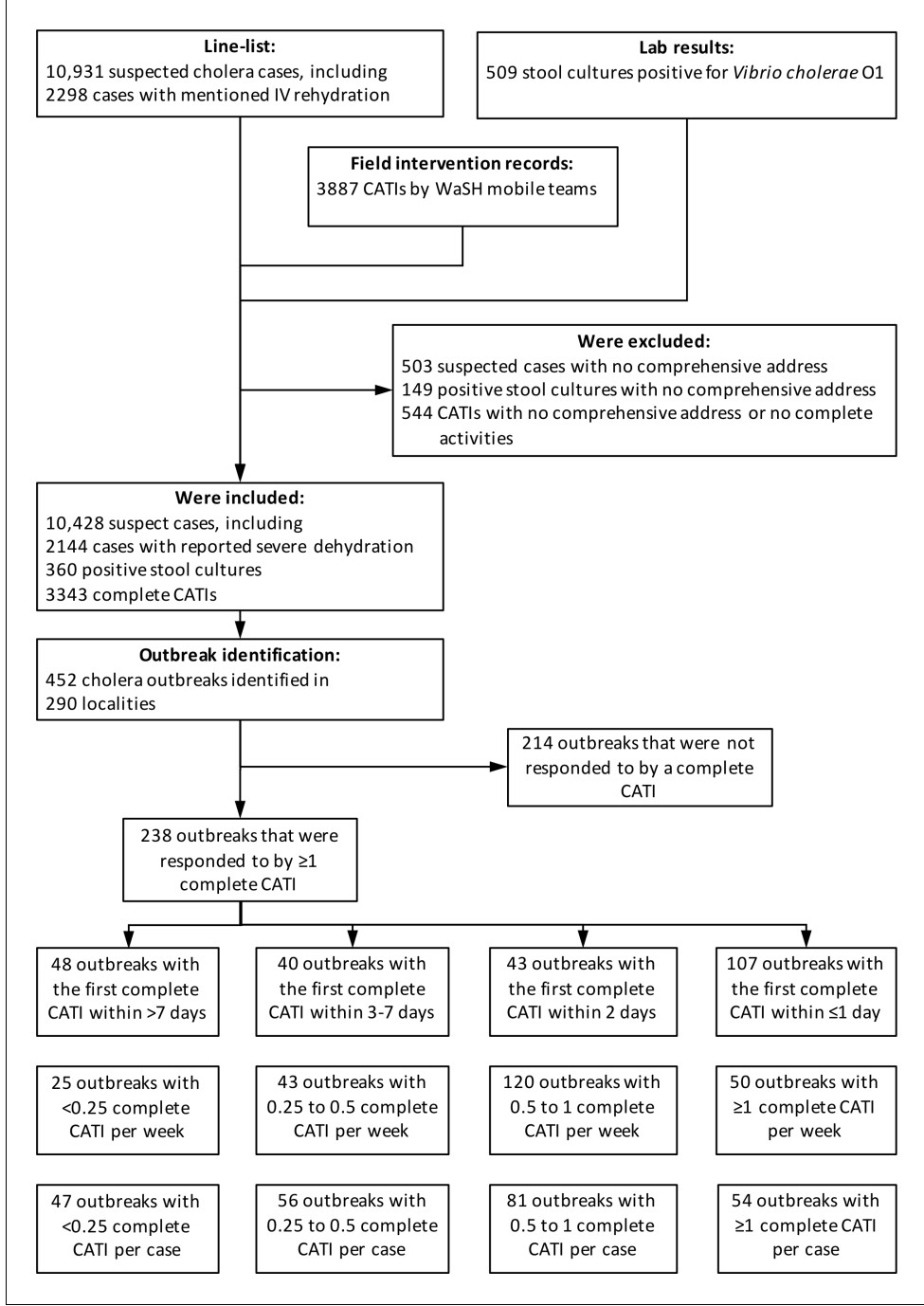

**Figure 2.** Identification of outbreaks and stratification of outbreaks according to response promptness and response intensity.

The online version of this article includes the following source data for figure 2:

**Source data 1.** Line-listing of suspected cholera cases, lab results of stool cultures for *Vibrio cholerae* O1 and list of case-area targeted interventions against cholera in the Centre department of Haiti between January 2015 and December 2017.

with prompter responses exhibited significantly more positive cultures during the first three days than outbreaks with delayed responses. None of the other covariates were significantly associated with response intensity (*Table 1*).

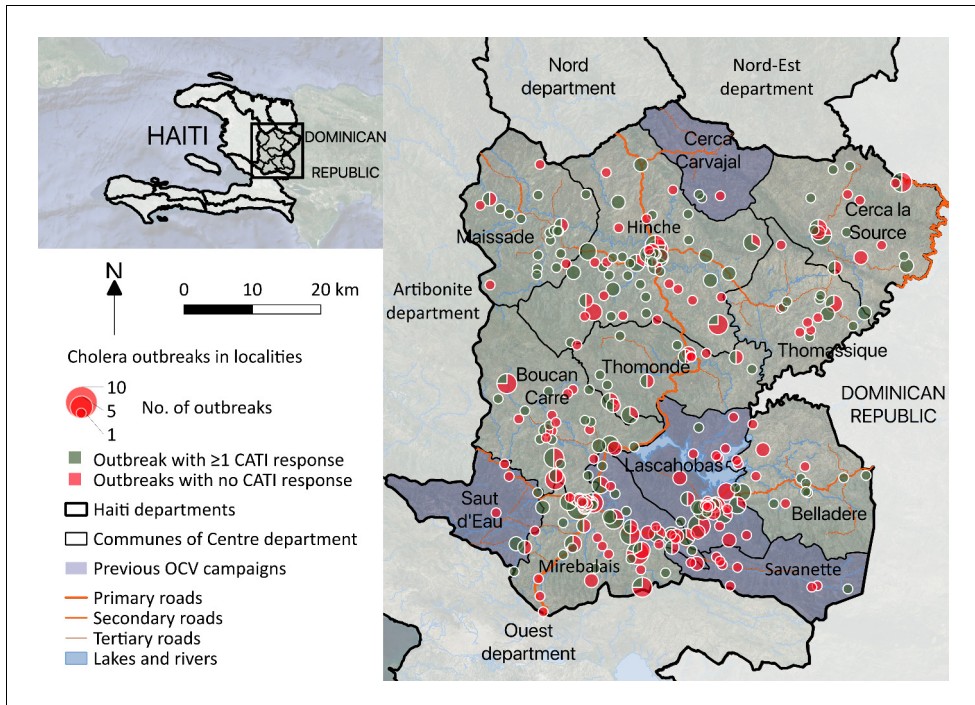

**Figure 3.** Cholera outbreaks in the Centre department, Haiti, between January 2015 and December 2017: spatial distribution and number of identified outbreaks (size of pie charts); proportion of outbreaks that were and were not responded with at least one complete CATI (angle of green and red slices, respectively).

Baseline characteristics of outbreaks and comparisons between the four classes of response intensity are presented in *Table 2*. The numbers of complete CATIs per week and per case (response intensity) significantly improved during the six semesters of the study. Outbreaks receiving more CATIs per case exhibited significantly fewer cases during the first three days than outbreaks receiving less intense responses (*Table 2*).

## CATI effectiveness according to the response promptness

There was a positive association between the time to the first complete CATI after outbreak onset, and the number of cases recorded from the fourth day of the outbreak (*Figure 4A*, *Table 3*). Consequently, the prompter the response, the higher the CATI effectiveness on the reduction of outbreak size (*Table 3*). Compared to a first complete CATI > 7 days after outbreak onset, the crude effectiveness of a first complete CATI $\leq$ 1 day ($cCE_1$) was 83% (95% CI, 71 to 90), and after adjusting for potential confounders ($aCE_1$), 76% (59 to 86).

Similarly, there was a positive association between the time to the first complete CATI after outbreak onset, and the duration of outbreaks (*Figure 4B*, *Table 4*). Consequently, the prompter the response, the higher the CATI effectiveness on the reduction of outbreak duration (*Table 4*). Compared to a first complete CATI > 7 days after outbreak onset, the crude effectiveness of a first complete CATI $\leq$ 1 day ($cCE_2$) was 59% (36 to 74), and after adjusting for potential confounders, ($aCE_2$) 61% (41 to 75).

## CATI effectiveness according to the response intensity

In addition, there was a negative association between the number of complete CATIs per week of outbreak, and the number of cases recorded from the fourth day of outbreak (*Figure 4C*, *Table 5*). Consequently, the more intense the response, the significantly higher the CATI effectiveness was estimated to be on the reduction of outbreak size (*Table 5*). Compared to a number of complete CATIs < 0.25 per week, the crude effectiveness of a number of complete CATIs $\geq$ 1 per week ($cCE_3$) was 74% (95% CI, 44 to 88), and after adjusting for potential confounders ($aCE_3$), 59% (11 to 81).

**Table 2.** Baseline characteristics of outbreaks that were responded to, according to the response intensity (number of complete case-area targeted interventions per week or per case).

| | No. of complete CATIs per week | | | | Comparison between classes of CATIs per week | |
|---|---|---|---|---|---|---|
| | <0.25 | 0.25 to 0.5 | 0.5 to 1 | ≥1 | OR (95% CI)[†] | p-value[†] |
| No. of outbreaks | 25 (11%) | 43 (18%) | 120 (50%) | 50 (21%) | | |
| Semester since January 2015 | | | | | 1.14 (1.03 to 1.25) | 0.0111[*] |
| Population density, median (IQR; inhab./km$^2$) | 3.9 (11.8) | 3.4 (3.8) | 3.4 (12) | 3.7 (11.8) | 1 (1 to 1.01) | 0.4093 |
| Travel time to the nearest town, median (IQR; minutes) | 30 (26.8) | 33.5 (41.9) | 22.1 (28) | 25.4 (34.2) | 1 (1 to 1.01) | 0.8379 |
| Accumulated incidence between 2010 and 2014, median (IQR; per 1000 inhab) | 125.8 (250.9) | 99.5 (97.8) | 103.8 (49) | 103.8 (79.6) | 0.64 (0.37 to 1.1) | 0.1037 |
| Coverage of OCV campaigns between 2012 and 2014, median (IQR; %) [mean, SD] | 0% (86) [43%, 44] | 0% (0) [11%, 29] | 0% (0) [19%, 36] | 0% (86) [25%, 40] | 1.03 (0.74 to 1.44) | 0.8464 |
| Previous cases in the same locality during the study, median (IQR; no. per year) | 10.1 (9.4) | 6 (8.8) | 4 (9) | 5.9 (11.7) | 1.01 (0.99–1.02) | 0.5011 |
| Previous complete CATIs in the same locality during the study, median (IQR; no. per year) | 0.7 (2.1) | 0.4 (2.5) | 1 (2.7) | 1.5 (3.2) | 1.04 (1–1.08) | 0.0763 |
| Daily rainfall during outbreak, median (IQR; mm) | 12 (4) | 8 (11.5) | 6.6 (16.3) | 6.2 (11.1) | 0.99 (0.98 to 1.01) | 0.331 |
| No. of cases during the first 3 daysof outbreak, median (IQR) [mean, SD] | 2 (0) [3.1, 2.7] | 2 (1) [2.7, 1.5] | 2 (0) [2.3, 1.4] | 3 (2) [3.4, 2.5] | 0.81 (0.71 to 0.93) | 0.3806 |
| No. of positive culture during the first 3 daysof outbreak, median (IQR) [mean, SD] | 0 (0) [0.2, 0.6] | 0 (0) [0.2, 0.4] | 0 (1) [0.4, 0.8] | 0 (0) [0.2, 0.6] | 1.03 (0.85 to 1.25) | 0.7569 |
| | No. of complete CATIs per case | | | | Comparison between classes of CATIs per case | |
| | <0.25 | 0.25 to 0.5 | 0.5 to 1 | ≥1 | OR (95% CI)[†] | p-value[†] |
| No. of outbreaks | 47 (20%) | 56 (24%) | 81 (34%) | 54 (23%) | | |
| Semester since January 2015 | | | | | 1.24 (1.13 to 1.37) | <0.0001[*] |
| Population density, median (IQR; inhab./km2) | 3 (5) | 4.2 (14.7) | 3.3 (3.6) | 3.7 (12.9) | 1 (1 to 1.01) | 0.468 |
| Travel time to the nearest town, median (IQR; minutes) | 31.2 (42.1) | 17.3 (44.4) | 25.5 (27.3) | 18.9 (23.7) | 1 (0.99 to 1) | 0.344 |
| Accumulated incidence between 2010 and 2014, median (IQR; per 1000 inhabitants) | 125.8 (77.5) | 103.8 (93.5) | 103.8 (64.6) | 103.8 (43.3) | 1.01 (0.42 to 2.43) | 0.981 |
| Coverage of OCV campaigns between 2012 and 2014, median (IQR; %) [mean, SD] | 0% (86) [30%, 42] | 0% (0) [13%, 31] | 0% (0) [21%, 38] | 0% (0) [22%, 38] | 0.96 (0.6 to 1.54) | 0.881 |

*Table 2 continued on next page*

Table 2 continued

| | No. of complete CATIs per week | | | | Comparison between classes of CATIs per week | |
|---|---|---|---|---|---|---|
| | <0.25 | 0.25 to 0.5 | 0.5 to 1 | ≥1 | OR (95% CI)[†] | p-value[†] |
| Previous cases in the same locality during the study, median (IQR; no. per year) | 4.4 (9.9) | 8.4 (16.3) | 6.2 (7.9) | 3.5 (6.4) | 1 (0.98–1.02) | 0.7730 |
| Previous complete CATIs in the same locality during the study, median (IQR; no. per year) | 0 (1) | 1.4 (4.3) | 1.4 (2.7) | 1 (2.6) | 1.03 (0.97–1.08) | 0.3550 |
| Daily rainfall during outbreak, median (IQR; mm) | 12 (4.3) | 6.1 (13.7) | 5.3 (13.6) | 6.1 (13.7) | 1 (0.98 to 1.02) | 0.983 |
| No. of cases during the first 3 daysof outbreak, median (IQR) [mean, SD] | 3 (3) [4.3, 3.2] | 2 (1) [2.6, 1.1] | 2 (0) [2.5, 1.3] | 2 (1) [1.8, 0.8] | 0.81 (0.71 to 0.93) | 0.0019[*] |
| No. of positive culture during the first 3 daysof outbreak, median (IQR) [mean, SD] | 0 (0) [0.2, 0.9] | 0 (0) [0.3, 0.5] | 0 (0) [0.2, 0.6] | 0 (1) [0.6, 0.7] | 1.14 (0.92 to 1.42) | 0.232 |

CATI, case-area targeted intervention; IQR, interquartile range; SD, standard deviation; OR (95% CI), Odds ratio (95%-confidence interval).

[†]Univariate comparisons using generalized linear mixed models with CATIs/weeks ratio or CATIs/cases ratio as model outcome and a negative-binomial distribution.

[*]Significant p-value.

Similarly, there was a negative association between the number of complete CATIs per case, and the duration of outbreaks (*Figure 4D*, *Table 6*). Consequently, the more intense the response, the significantly higher the CATI effectiveness on the reduction of outbreak duration (*Table 6*). Compared to a number of complete CATIs < 0.25 per case, the crude effectiveness of a number of complete CATIs ≥ 1 per case ($cCE_4$) was 76% (95% CI, 54 to 88), and after adjusting for potential confounders ($aCE_4$), 73% (49 to 86).

Several sensitivity analyses using alternative definitions of cholera outbreak, alternative definitions of CATIs and alternative methods of covariate selection for adjustment yielded consistent estimates of CATI effectiveness according to response promptness and response intensity (Appendix 3).

## Effectiveness of antibiotic prophylaxis

Finally, stratified analyses showed that three estimates of CATI effectiveness out of four appeared higher in the subgroup of outbreaks that were only responded to by complete CATIs with antibiotic prophylaxis (ATB) than in the subgroup of outbreaks only responded to by complete CATIs that never included ATB (*Table 7*). More precisely, the adjusted effectiveness of a prompt response on outbreak size ($aCE_1$) was 63% (24 to 82) when all CATIs included antibiotic prophylaxis, and 39% (-38 to 73) when no CATI did. The adjusted effectiveness of a prompt response on outbreak duration ($aCE_2$) was 74% (43 to 88) when all CATIs included antibiotic prophylaxis, and 58% (11 to 80) when no CATI did. Similarly, the adjusted effectiveness of an intense response on outbreak duration ($aCE_4$) was 90% (72 to 96) when all CATIs included antibiotic prophylaxis, and 79% (46 to 92) when no CATI did. Conversely, the adjusted effectiveness of an intense response on outbreak size ($aCE_3$) was 62% (3 to 85) when all CATIs included antibiotic prophylaxis, and 76% (12 to 94) when no CATI did (*Table 7*).

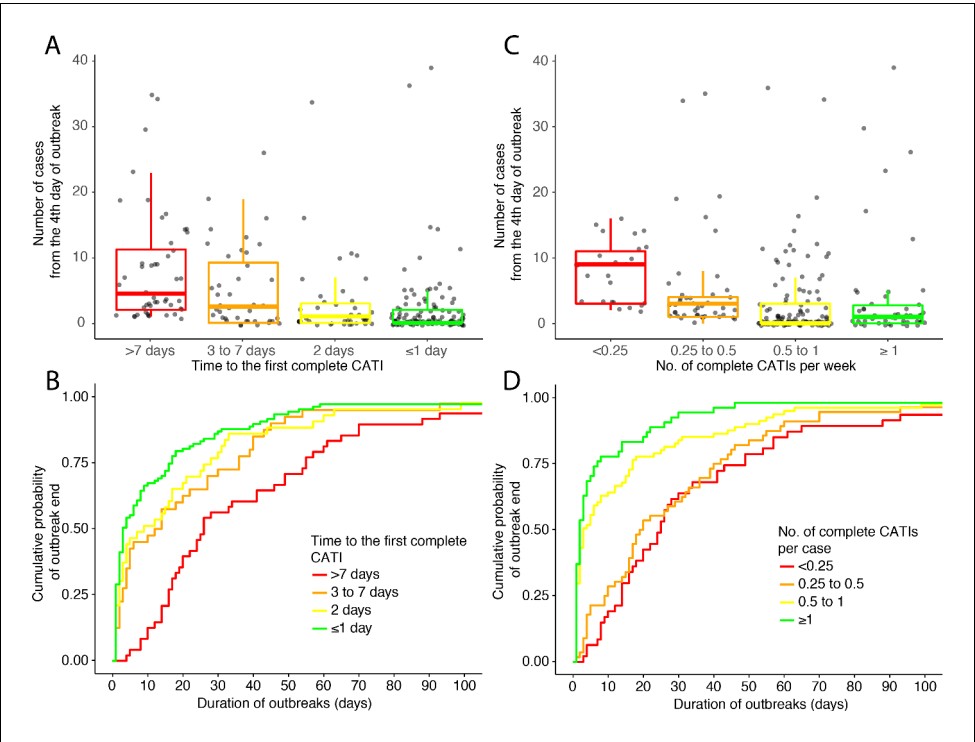

**Figure 4.** Outbreak outcome according to the class of response promptness. (**A and B**) and to the class of response intensity (**C and D**): (**A and C**) comparison of the outbreak size (number of suspected cholera cases from the fourth day of outbreak) and (**B and D**) Kaplan-Meier comparison of the outbreak duration (in days), according to the time to the first complete CATI (**A and B**), to the number of complete CATIs per week (**C**) and to the number of complete CATIs per case (**D**).

The online version of this article includes the following source data for figure 4:

**Source data 1.** Main characteristics of outbreak response and outcome: class of time to the first complete CATI; class of number of complete CATIs per week; class of number of complete CATIs per case; number of suspected cholera cases from the fourth day of outbreaks; duration of outbreaks.

## Discussion

Our quasi-experimental study, based on epidemiological and intervention records over three years in one administrative department of Haiti, showed that prompt and repeated response CATIs conducted by rapid response teams were significantly associated with shortening of cholera outbreaks and mitigating of outbreak case load. Of note, numerous suspected cholera outbreaks spontaneously ended before any response could be conducted. But when taking into account this significant confounding by indication (*Remschmidt et al., 2015*), the prompter the first complete CATI was implemented and the more complete CATIs were conducted, the fewer cases were recorded and the shorter the outbreak lasted.

While many mild suspected outbreaks may spontaneously end without any response intervention, prompt and repeated CATIs appear difficult to sustain during the largest outbreaks. As suggested by the slow increase in the number of CATIs observed during the study period, CATIs may be logistically complex to implement, and response teams can become overwhelmed when they try to simultaneously address a large number of cases (*Finger et al., 2018*; *Rebaudet et al., 2019a*). Such strategy certainly is most relevant at the beginning of epidemics, or during trough periods or tails of epidemics (*Finger et al., 2018*; *Rebaudet et al., 2013*).

Our study comes with a number of limitations. Because CATIs were not randomized, response effectiveness may have been biased by unmeasured confounders. As we observed a significant confounding by indication on the probability for an outbreak to receive a CATI response, we limited our analysis on outbreaks that were responded to. We subsequently did not observe any consistent residual difference of initial severity between classes of response promptness and response intensity.

**Table 3.** CATI effectiveness ($CE_1$) of the response promptness (time to the first complete CATI) on outbreak size (number of cases from the fourth day of outbreak).

| | | No. of cases from the 4th day of outbreak | Crude estimate of CATI effectiveness ($cCE_1$)[†] | | Adjusted estimate of CATI effectiveness ($aCE_1$)[‡] | |
|---|---|---|---|---|---|---|
| | N | Median (IQR) | % (95% CI) | p-value | % (95% CI) | p-value |
| Time to the first complete CATI | | | | | | |
| >7 days | 48 | 4.5 (9.25) | Ref | Ref | Ref | Ref |
| 3 to 7 days | 40 | 2.5 (9.25) | 49% (6 to 72) | 0.0318[*] | 50% (9 to 72) | 0.0222[*] |
| 2 days | 43 | 1 (3) | 76% (55 to 87) | <0.0001[*] | 68% (40 to 83) | 0.0004[*] |
| ≤1 day | 107 | 0 (2) | 83% (71 to 90) | <0.0001[*] | 76% (59 to 86) | <0.0001[*] |

CATI, case-area targeted intervention; IQR, interquartile range.

[*]Significant p-value.

[†]Crude CATI effectiveness ($cCE_1$) was estimated on the No. of cases from the fourth day of outbreak, using generalized linear mixed models with a negative-binomial distribution, as (1 − Incidence ratio).

[‡]Estimates of CATI effectiveness ($aCE_1$) were adjusted according to covariates for which p-values were less than 0.25 at the initial univariate step (**Table 1**): number of positive cultures during the first 3 days of outbreak, population density, accumulated case incidence between 2010 and 2014, coverage of OCV campaigns between 2012 and 2014 and semester.

However, our models were adjusted for potential confounders and took into account the heterogeneity between localities. This quasi-experimental study was also stratified on response promptness and on response intensity, which yielded consistent response effectiveness estimates (*Shadish et al., 2002*).

Analyses may also have been biased by missing epidemiological data. Indeed, some patients do not seek care, even when they experience severe dehydration. Besides, stool sampling for confirmation culture was not systematic, which certainly led us to overlook several authentic outbreaks and mis-select clusters of non-cholera diarrhoeas. It may have led us to misdate several outbreak onset

**Table 4.** CATI effectiveness ($CE_2$) of the response promptness (time to the first complete CATI) on outbreak duration (in days).

| | | Duration of outbreak | Crude estimate of CATI effectiveness ($cCE_2$)[†] | | Adjusted estimate of CATI effectiveness ($aCE_2$)[‡] | |
|---|---|---|---|---|---|---|
| | N | Median (IQR; days) | % (95% CI) | p-value | % (95% CI) | p-value |
| Time to the first complete CATI | | | | | | |
| >7 days | 48 | 26 (39) | Ref | Ref | Ref | Ref |
| 3 to 7 days | 40 | 13 (33) | 45% (17 to 64) | 0.0046[*] | 53% (29 to 69) | 0.0004[*] |
| 2 days | 43 | 9 (25) | 37% (−6 to 62) | 0.0810 | 27% (−22 to 56) | 0.2322 |
| ≤1 day | 107 | 3 (15.5) | 59% (36 to 74) | <0.0001[*] | 61% (41 to 75) | <0.0001[*] |

CATI, case-area targeted intervention; IQR, interquartile range.

[*]Significant p-value.

[†]Crude CATI effectiveness ($cCE_2$) was estimated on the duration of outbreak, using Cox models for Andersen-Gill counting process (AG-CP), as (1−1/hazard ratio).

[‡]Estimates of CATI effectiveness ($aCE_2$) were adjusted according to covariates for which p-values were less than 0.25 at the initial univariate step (**Table 1**): number of positive cultures during the first 3 days of outbreak, population density, accumulated case incidence between 2010 and 2014, coverage of OCV campaigns between 2012 and 2014 and semester.

**Table 5.** CATI effectiveness ($CE_3$) of the response intensity (number of complete CATIs per week) on outbreak size (number of cases from the fourth day of outbreak).

| | N | No. of cases after the 4th day of outbreak Median (IQR) | Crude estimate of CATI effectiveness ($cCE_3$)[†] % (95% CI) | p-value | Adjusted estimate of CATI effectiveness ($aCE_3$)[‡] % (95% CI) | p-value |
|---|---|---|---|---|---|---|
| **No. of complete CATIs per week** | | | | | | |
| <0.25 | 25 | 9 (8) | Ref | Ref | Ref | Ref |
| 0.25 to 0.5 | 43 | 3 (3) | 55% (1 to 79) | 0.0457[*] | 45% (−17 to 74) | 0.1206 |
| 0.5–1 | 120 | 0 (3) | 79% (59 to 89) | <0.0001[*] | 70% (42 to 84) | 0.0003[*] |
| ≥1 | 50 | 1 (2.75) | 74% (44 to 88) | 0.0006[*] | 59% (11 to 81) | 0.0235[*] |

CATI, case-area targeted intervention; IQR, interquartile range.

[*]Significant p-value.

[†]Crude CATI effectiveness ($cCE_3$) was estimated on the No. of cases from the fourth day of outbreak, using generalized linear mixed models with a negative-binomial distribution, as (1 − Incidence ratio).

[‡]Estimates of CATI effectiveness ($aCE_3$) were adjusted according to covariates for which p-values were less than 0.25 at the initial univariate step (**Table 2**): accumulated case incidence between 2010 and 2014, and semester.

and outbreak end. Depending on the differential distribution of these potential biases among classes of response promptness and intensity, these limits could have led to over- or under-estimation of the effectiveness of prompt and intense CATIs. Nevertheless, our outbreak definition aimed to deal with those missing data and be specific in order to analyse CATI effectiveness on definite outbreaks. Like for many diseases, no standardized cholera outbreak criteria exists, and several definitions may be more or less suitable depending on interventions and analyses objectives (**Brady et al., 2015**). Our retrospectively defined outbreaks may be an approximate unit of analysis in terms of space, time and population, which may also have biased effectiveness results. We therefore conducted a sensitivity analysis using alternative definitions, including systematically lab-confirmed cholera

**Table 6.** CATI effectiveness ($CE_4$) of the response intensity (number of complete CATIs per case) on outbreak duration (in days).

| | N | Duration of outbreak Median (IQR; days) | Crude estimate of CATI effectiveness ($cCE_4$)[†] % (95% CI) | p-value | Adjusted estimate of CATI effectiveness ($aCE_4$)[‡] % (95% CI) | p-value |
|---|---|---|---|---|---|---|
| **No. of complete CATIs per case** | | | | | | |
| <0.25 | 47 | 25 (32) | Ref | Ref | Ref | Ref |
| 0.25 to 0.5 | 56 | 19.5 (30.75) | 8% (−35 to 37) | 0.6738 | 1% (−45 to 32) | 0.9759 |
| 0.5 to 1 | 81 | 3 (16) | 59% (35 to 75) | 0.0002[*] | 57% (30 to 74) | 0.0007[*] |
| ≥1 | 54 | 2 (5.75) | 76% (54 to 88) | <0.0001[*] | 73% (49 to 86) | <0.0001[*] |

CATI, case-area targeted intervention; IQR, interquartile range.

[*]Significant p-value.

[†]Crude CATI effectiveness ($cCE_4$) was estimated on the duration of outbreak, using Cox models for Andersen-Gill counting process (AG-CP), as (1−1/hazard ratio).

[‡]Estimates of CATI effectiveness ($aCE_4$) were adjusted according to covariates for which p-values were less than 0.25 at the initial univariate step (**Table 2**): number of cases and number of positive cultures during the first 3 days of outbreak, yearly number of previous complete CATIs during the study, and semesters.

**Table 7.** Effectiveness of complete CATIs stratified by antibiotic prophylaxis.

| Outbreak subgroup | All outbreaks responded to by any complete CATIs (*Tables 3–6*) | | Outbreaks only responded to by complete CATIs with ATB | | Outbreaks only responded to by complete CATIs without ATB | |
|---|---|---|---|---|---|---|
| No. of outbreaks that were responded to (%) | 238 (53%) | | 115 (25%) | | 78 (17%) | |
| | % (95% CI) | p-value | % (95% CI) | p-value | % (95% CI) | p-value |
| **CATI effectiveness according to the response promptness** | | | | | | |
| ≤1 day vs > 7 days adjusted estimate of CATI effectiveness on accumulated cases (aCE$_1$)[†] | 76% (59 to 86) | <0.0001[*] | 63% (24 to 82) | 0.007[*] | 39% (−38 to 73) | 0.2369 |
| ≤1 day vs > 7 days adjusted estimate of CATI effectiveness on outbreak duration (aCE$_2$)[‡] | 61% (41 to 75) | <0.0001[*] | 74% (43 to 88) | 0.0009[*] | 58% (11 to 80) | 0.0237[*] |
| **CATI effectiveness according to the response intensity** | | | | | | |
| ≥1 vs<0.25 completeCATIs per week adjusted estimate of CATI effectivenesson accumulated cases (aCE$_3$)[$] | 59% (11 to 81) | 0.0235[*] | 62% (3 to 85) | 0.042 | 76% (12 to 94) | 0.0312 |
| ≥1 vs<0.25 completeCATIs per case adjusted estimate of CATI effectivenesson outbreak duration (aCE$_4$)[£] | 73% (49 to 86) | <0.0001[*] | 90% (72 to 96) | <0.0001[*] | 79% (46 to 92) | 0.0012[*] |

CATI, case-area targeted intervention.

ATB, antibiotic prophylaxis.

[†]Estimates of CATI effectiveness (aCE$_1$) were adjusted according to covariates for which p-values were less than 0.25 at the initial univariate step (**Table 1**): number of positive cultures during the first 3 days of outbreak, population density, accumulated case incidence between 2010 and 2014, coverage of OCV campaigns between 2012 and 2014 and semester.

[‡]Estimates of CATI effectiveness (aCE$_2$) were adjusted according to covariates for which p-values were less than 0.25 at the initial univariate step (**Table 1**): number of positive cultures during the first 3 days of outbreak, population density, accumulated case incidence between 2010 and 2014, coverage of OCV campaigns between 2012 and 2014 and semester.

[$]Estimates of CATI effectiveness (aCE$_3$) were adjusted according to covariates for which p-values were less than 0.25 at the initial univariate step (**Table 2**): accumulated case incidence between 2010 and 2014, and semester.

[£]Estimates of CATI effectiveness (aCE$_4$) were adjusted according to covariates for which p-values were less than 0.25 at the initial univariate step (**Table 2**): number of cases and number of positive cultures during the first 3 days of outbreak, yearly number of previous complete CATIs during the study, and semesters.

[*]significant after Bonferroni correction.

outbreaks, which showed consistent and robust estimates (Appendix 3.1). We also used mixed models in order to take into account heterogeneity between localities in the random effect (*Berridge and Crouchley, 2011*). Additional CATI effectiveness studies at the household and at the administrative commune levels are underway in Haiti.

Our study analysed 3887 CATIs prospectively notified by rapid response teams to UNICEF. But some additional CATIs may have been omitted, while other CATIs remained unrecorded because they were implemented by the EMIRA alone. Nevertheless, many of their respective CATIs actually overlapped, and we thus believe our response database to be reasonably exhaustive. Conversely, only 16% of complete CATIs were conducted in a locality experiencing a current outbreak. The remaining CATIs were implemented in response to sporadic cases that did not meet outbreak definition criteria, as illustrated by much higher rates with less stringent outbreak definitions (Appendix 3.1). Sporadic CATIs may have prevented, delayed or attenuated the emergence of outbreaks. They may also be associated with the propensity of future outbreak response. We thus included the frequency of previous complete CATIs in our analysis but found no significant association with response promptness or intensity.

Our study aimed to assess the overall effectiveness of a CATI strategy. It neither aimed to estimate the respective effectiveness of each response components, nor the optimal radius of intervention, which would warrant dedicated field studies comparing different types of interventions. We thus chose a conservative definition of complete CATIs and performed a sensitivity analysis with alternative CATI definitions that exhibited consistent results (Appendix 3.2). Because nearly all CATIs included house decontamination, education and chlorine distribution, stratified analyses on these activities were not possible. However, three effectiveness estimates out of four appeared higher when all CATIs included antibiotic prophylaxis than when no CATI did. Several trials have also suggested that chemoprophylaxis has a protective effect among household contacts of people with cholera (*Reveiz et al., 2011*), and a micro-simulation model suggested that administration of antibiotics in CATIs could effectively avert secondary cases (*Finger et al., 2018*). But considering the risk of resistance selection (*Mhalu et al., 1979*; *Dromigny et al., 2002*), the selected distribution of antibiotic prophylaxis to close contacts is usually not recommended (*Global Task Force on Cholera Control (GTFCC), 2018*) and must, at the minimum, be used with caution and close monitoring of antibiotic susceptibility. In Haiti, all clinical *V. cholerae* O1 isolates have remained susceptible to doxycycline between 2013 and 2019 (Haitian Ministry of Public Health and Population, MSPP). As suggested by previous field or modelling studies (*Ali et al., 2016*; *Parker et al., 2017b*; *Parker et al., 2017a*; *Finger et al., 2018*), adding the administration of a single-dose OCV during CATIs could be an effective, but likely logistically complex, strategy.

Overall, our results suggest that case-area targeted interventions are significantly effective to mitigate and shorten local cholera outbreaks. Household water treatment, sanitation and hygiene promotion, as well as antibiotic prophylaxis theoretically prevent both human-to-human and environment-to-human cholera transmission pathways. Regardless of their respective role, which has been much debated (*Morris, 2011*; *Kupferschmidt, 2017*; *Rebaudet et al., 2019b*), our results thus confirm the relevance of promoting rapid response teams as a key component of the new global strategy for cholera control (*Global Task Force on Cholera Control (GTFCC), 2017*; *Anon, 2017*). Such findings need to be replicated in other settings and at other spatial and time scales. It will be critical to understand where CATIs should be prioritized, which radius is optimal, and which intervention components are most effective.

## Materials and methods

### Study design, setting and cholera surveillance

To assess CATI effectiveness, we conducted a retrospective observational study, which compared the outcome of cholera outbreaks according to the promptness or intensity of response CATIs. This corresponded to a quasi-experimental study using a post test-only design with stratified groups (*Shadish et al., 2002*). The study was conducted from January 1, 2015, to December 31, 2017 in the Centre department, one of the 10 administrative districts of Haiti. Centre department covers an area of 3487 km$^2$, with an altitude ranging from 69 m to 1959 m, and is administratively subdivided in 12 communes. In 2015, the Centre population was estimated to be 746,236 inhabitants, including 20% living in urban neighbourhoods, and 80% in numerous rural settlements (*Institut Haitien de Statistique et d'Informatique (IHSI), 2015*). For the purpose of this study, we designate urban neighbourhoods and rural settlements as 'localities'.

In 2015–2017, 17 cholera treatment centres, cholera treatment units and acute diarrhoea treatment centres officially treated and recorded suspected cholera cases and associated deaths to the MSPP. A probable suspected cholera case was defined as a patient who develops acute watery diarrhoea with or without vomiting. Daily cases and deaths tolls aged <or $\geq$ five years old were separately notified to the department health directorates. From 2014, the health directorate of the Centre department established a line-listing of all suspected cholera cases, mentioning sex, age, date of admission, address and use of IV rehydration (a surrogate for severe dehydration). Finally, routine bacteriological confirmation of a subset of suspected cholera cases was performed at the National Laboratory of Public Health (LNSP) in Port-au-Prince Metropolitan Area, using stool sampling with Carry-Blair transport medium and standard culture and phenotyping methods (*Centers for Disease Control and Prevention (CDC), 1999*).

## Procedures: rapid case-area targeted interventions (CATIs)

From July-2013, the nationwide case-area targeted rapid response strategy to eliminate cholera in Haiti was laboriously but increasingly implemented throughout the country (*Rebaudet et al., 2019a*). In the Centre department between 2015 and 2017, UNICEF established a partnership with Zanmi Lasante, Oxfam, ACTED and IFRC (International Federation of Red Cross and Red Crescent), four NGOs that hired WaSH rapid response teams composed of local Haitian staff. MSPP also established its own teams called EMIRAs, which included healthcare workers (nurses, auxiliary nurses). Staff of the NGO rapid response teams and EMIRA worked together and deployed mixed teams, which were requested to respond to every suspected cholera case or death within 48 hr after admission at healthcare facility. For this purpose, rapid response teams were encouraged to get epidemiological cholera data on a daily basis from departmental health directorates and treatment centres (*Rebaudet et al., 2019a*). The core methodology of response CATIs had been established with the MSPP and its partners and included: (i) door-to-door visits to affected families and their neighbours (minimum five households depending on the local geography), who were proposed house decontamination by chlorine spraying of latrines and other potentially contaminated surfaces; (ii) on-site organization of education sessions about cholera and hygiene promotion; (iii) and distribution of one cholera kit per household (composed of five soaps, five sachets of ORS, and chlorine tablets [80 Aquatabs33 mg in urban settings or 150 Aquatabs in rural areas]). EMIRA staff also provided (iv) prophylactic antibiotics to contacts living in the same house as cholera cases with one dose of doxycycline 300 mg for non-pregnant adults only. When appropriate, rapid response teams also: (v) established manual bucket chlorination at drinking water collection points during one or more weeks, by hiring and instructing local volunteers; (vi) chlorinated water supply systems and reported potential malfunctions to DINEPA; (vii) supervised safe funeral practices for cholera casualties; and (viii) provided primary care to cholera cases found in the community. CATIs were prospectively documented and transmitted by WaSH rapid response teams to UNICEF with date, location (*i.e.,* commune, communal section, locality) and implemented activities, including specific activities of embedded EMIRA staff.

Response CATIs were defined as complete if rapid response teams reported at least education, decontamination and distribution of chlorine tablets. A sensitivity analysis of CATI effectiveness estimates using alternative CATI definitions is provided in Appendix 3.2.

## Outbreaks identification and characterization

In order to identify cholera outbreaks, we first cleaned the anonymised case line-listing provided by the health directorate of the Centre department, the anonymised stool culture database provided by the LNSP and the response database provided by UNICEF. We manually corrected date errors and duplicates. Using repeated field investigations, GPS coordinates provided by rapid response teams, and several geographic repositories (http://ihsi.ht/publication_cd_atlas.htm, https://www.indexmundi.com/zp/ha/, https://www.openstreetmap.org/, https://www.google.fr/maps, accessed Jul 1, 2019), we corrected case, culture and response addresses with unified and geolocated locality names. We included every suspected case, every stool culture positive for *V. cholerae* O1 and every complete CATI of a WaSH rapid response team reported in the Centre department between January 2015 and December 2017.

To assess response effectiveness, we needed to escape the double pitfall of an overly restrictive definition of outbreaks, for example by requiring a bacteriological documentation for each suspected case and, on the contrary, of an unspecific definition, in which a large number of non-cholera diarrhoea cases would have been included. In addition, we had to deal with the fact that some patients with a positive culture were missing in the line-listing. Considering the median and the maximum O1-serogroup cholera incubation period are about 1.5 and 7 days, respectively (*Azman et al., 2013*), we thus defined outbreaks by the occurrence of at least two suspected cholera cases with at least one severely dehydrated case or positive culture, within the same locality, during a three-day time window, and after a refractory period of at least 21 days with no case. Outbreak onset was defined as the date of the first suspected case or positive culture, and outbreak end as the date of the last case or positive culture before a refractory period of at least 21 days. We conducted a sensitivity analysis using alternative outbreak definitions (Appendix 3.1).

For each identified outbreak, we then counted the numbers of cases and positive culture during the first three days as surrogates of initial severity. With a median incubation period of 1.5 days (*Azman et al., 2013*), we considered that a response – even a prompt one – would have little impact on the occurrence of additional cases during the two days following detection of the first case. Using a geographic information system (GIS), we extracted locality characteristics such as median population density (*Sorichetta et al., 2015*) and travel time to the nearest town (*Weiss et al., 2018*), using 1000 m radius buffer zones. Because cholera transmission and CATI response against cholera in Haiti were found to be influenced by rainfall (*Eisenberg et al., 2013*; *Rebaudet et al., 2019a*), we obtained NASA satellite estimates of daily-accumulated rainfall (TRMM_3B42_daily v7, area-averaged with 0.25° x 0.25° accuracy) (https://giovanni.gsfc.nasa.gov/giovanni/, accessed Jul 1, 2019). We gathered vaccine coverage of OCV campaigns conducted between 2012 and 2014, and accumulated incidence rates of suspected cholera cases between 2010 and 2014 (Haitian Ministry of Public Health and Population, MSPP), as surrogates of the population immunity against cholera. In order to better take into account the propensity of localities to experience outbreaks and receive response CATIs, we also counted the number of previous cases per year and the number of previous complete CATIs per year in the same locality since the beginning of the study. To take into account the possible variation of CATI implementation and effectiveness over time, we divided the three-year study period into six semesters (first and last six months of every year).

We then considered that outbreaks were responded to if at least one complete CATI was implemented within seven days after the last recorded case of the outbreak. In order to characterize the response promptness in this subgroup of outbreaks, we first counted the number of days between outbreak onset and the first complete CATI, and split outbreaks that were responded to between four classes of response promptness: >7 days, 3 to 7 days, 2 days and ≤1 day. In order to characterize the response intensity in the subgroup of outbreaks that were responded to, we also counted the number of complete CATIs per outbreak, divided this number by the outbreak duration (in week), and split outbreaks that were responded to between four classes of response intensity: <0.25, 0.25 to 0.5, 0.5 to 1 and ≥1 CATIs per week. We also divided the number of complete CATIs per outbreak by the number of accumulated cases per outbreak, and split outbreaks that were responded to between four classes of response intensity: <0.25, 0.25 to 0.5, 0.5 to 1 and ≥1 CATIs per case. Finally, we calculated two surrogates of outbreak outcome: the number of accumulated suspected cases from the fourth day of outbreak (outbreak size), and the number of days between the first and the last reported case or culture (outbreak duration). A sensitivity analysis of CATI effectiveness using alternative response time windows and categories is provided in Appendix 3: Sensitivity analyses of CATI effectiveness.

## Statistical analysis
### Analysis of confounders

The assessment of a possible confounding by indication is detailed in Appendix 2 (*Remschmidt et al., 2015*). We found that CATI response was more likely in more severe outbreaks. To handle this major bias, we therefore assessed CATI effectiveness (CE) by analysing the outcome of outbreaks that were responded to, according to the response promptness and according to response intensity. In two separate analyses, we compared two endpoints between the four classes of response promptness and between the four classes of response intensity (exposure): the number of cases from the fourth day of outbreak (CE represented the proportion of averted cases); and the outbreak duration (CE represented the proportion of averted days).

As an initial univariate step, we looked for possible confounders among baseline outbreak characteristics of response groups. First, each possible confounder was modelled as an independent variable, and time to the first complete CATI as a recurrent time-to-event outcome, using Cox survival models for Andersen-Gill counting process (AG-CP). This AG-CP survival model was chosen to take into account the correlated repetitions of outbreaks within localities (*Andersen and Gill, 1982*). Each possible confounder was also modelled as a fixed effect variable, the number of CATIs per week or the number of CATIs per case as dependent variables, and localities as a random effect, using generalized linear mixed models (GLMMs) with a negative-binomial distribution. The mixed model approach aimed to take into account the homogeneous pattern within localities, and the negative-binomial distribution to take into account overdispersion (*Berridge and Crouchley, 2011*).

## CATI effectiveness according to response promptness

The first evaluation of CATI effectiveness ($CE_1$) was then performed by comparing the outbreak size (number of cases from the fourth day of outbreak) between the four classes of response promptness (time to the first complete CATI). For this, we used GLMMs with cases from the fourth day of outbreak as a dependent variable, localities as a random effect, and a negative-binomial distribution (*Berridge and Crouchley, 2011*). For each class of response promptness, we estimated the crude CATI effectiveness ($cCE_1$) as: 1 – Incidence ratio. We then obtained adjusted estimates of CATI effectiveness ($aCE_1$) by adjusting for confounders for which p-values were less than 0.25 at the initial univariate step (*Mickey and Greenland, 1989*). A sensitivity analysis of CATI effectiveness using alternative methods of covariate selection is provided in Appendix 3.3.

A second evaluation of CATI effectiveness ($CE_2$) was performed by comparing the outbreak duration between the four classes of response promptness, using survival analyses censoring outbreak extinction. We assessed time-to-event by Kaplan-Meier analysis to illustrate the cumulative probability of outbreak end between the different response promptness classes. In order to estimate CATI effectiveness according to response promptness and take into account the correlated repetitions of outbreaks within localities, we then fitted Andersen-Gill (AG-CP) survival models (*Andersen and Gill, 1982*). For each class of response promptness, we estimated the crude CATI effectiveness ($cCE_2$) as: 1 – (1/Hazard ratio). We then obtained adjusted estimates of CATI effectiveness ($aCE_2$) using the same methodology.

## CATI effectiveness according to response intensity

We then estimated the effectiveness of the response intensity, by comparing the outbreak size or duration between different classes of response intensity, using the same methodology as for the effectiveness according to response promptness. In order to avoid that cases or duration be included both within outcome and exposure variables, we approximated response intensity by the number of complete CATIs per week ratio when comparing the number of cases accumulated from the fourth day of outbreak ($CE_3$). Conversely, we used the number of complete CATIs per case ratio when comparing the duration of outbreak ($CE_4$).

For all effectiveness analyses, a p-value of less than 0.05 (two-sided) was considered to indicate statistical significance.

## Effectiveness of antibiotic prophylaxis

In order to assess the effectiveness of antibiotic prophylaxis, we conducted similar comparisons of outbreak size using GLMMs or outbreak duration using Andersen-Gill (AG-CP) survival models according to the response promptness or to the response intensity, stratified by whether all complete CATIs or none of the complete CATIs included antibiotic prophylaxis. We adjusted estimates of CATI effectiveness for the same confounders as in previous analyses. Using a Bonferroni correction for multiple comparisons, a p-value of less than 0.025 (two-sided) was considered to indicate statistical significance.

## Software

The GIS and the map were done using QGIS software v3.03 and layers obtained from Haiti Centre National de l'Information Géospatiale (CNIGS) (http://cnigs.ht/, accessed Jul 1, 2019). Analyses and graphs were done using RStudio version 1.0.136 for Mac with R version 3.4.2 and the {ggplot2}, {lme4}, {survival} and {survminer} packages.

# Acknowledgements

We are grateful to the staff of MSPP, UNICEF, DINEPA and NGOs, who cared for patients, implemented and coordinated field responses, analysed stool cultures, gathered epidemiological and intervention data. We thank Bevan Hurley for editing the manuscript.

## Additional information

### Funding

| Funder | Author |
|---|---|
| UNICEF | Samuel Beaulieu<br>Gregory Bulit<br>Stanislas Rebaudet |
| Assistance Publique - Hôpitaux de Marseille | Stanislas Rebaudet |
| Assistance Publique - Hôpitaux de Paris | Renaud Piarroux |

The funders of this study (UNICEF, APHM, APHP) had staff (co-authors of this manuscript) who had a role in data collection, analyses and writing of the report. However, the funders had no role in study design, data collection and analysis, decision to publish, or preparation of the manuscript.

### Author contributions

Edwige Michel, Data curation, Formal analysis, Investigation; Jean Gaudart, Conceptualization, Software, Supervision, Validation, Methodology; Samuel Beaulieu, Gregory Bulit, Funding acquisition, Validation, Investigation; Martine Piarroux, Supervision, Investigation, Methodology; Jacques Boncy, Funding acquisition, Investigation, Project administration; Patrick Dely, Supervision, Funding acquisition, Project administration; Renaud Piarroux, Conceptualization, Supervision, Funding acquisition, Validation, Investigation, Methodology; Stanislas Rebaudet, Conceptualization, Data curation, Software, Formal analysis, Supervision, Funding acquisition, Validation, Investigation, Visualization, Methodology, Project administration

### Author ORCIDs

Jean Gaudart (ID) http://orcid.org/0000-0001-9006-5729
Stanislas Rebaudet (ID) https://orcid.org/0000-0001-5099-1947

### Ethics

Human subjects: All analyses retrospectively included routinely collected cholera surveillance and control data. The study protocol received authorization #1718-24 from the National Bioethics Committee of Haiti Ministry of Health (MSPP). The study only analysed anonymised data. Informed consent from patients and from people who benefited from a response intervention was therefore not required for this study.

### Decision letter and Author response

Decision letter https://doi.org/10.7554/eLife.50243.sa1
Author response https://doi.org/10.7554/eLife.50243.sa2

## Additional files

### Supplementary files

• Source code 1. R code for statistical comparisons between outbreak groups and estimation of response effectiveness.

• Transparent reporting form

### Data availability

Data generated or analysed during this study are included in the manuscript and supporting files. Source data files have been provided for Figures 1 and 4.

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

## Appendix 1

### Data characteristics

The study was conducted from January 1, 2015, to December 31, 2017 in the Centre department, where the cholera started in October 2010 (*Piarroux et al., 2011*). Like the rest of the country, this administrative district experienced a massive epidemic in 2010–2011 (*Gaudart et al., 2013*; *Figure 1—figure supplement 1*). Incidence then gradually decreased from 2012 to 2014, but in the following years, the Centre department remained one of the most affected area of the country (*Rebaudet et al., 2019a*). In order to better guide the response CATI strategy, the health directorate of the Centre department established case line-listing from 2014. This gave us the opportunity to retrospectively assess the impact of these interventions at the level of localities between 2015 and 2017 (*Figure 1—figure supplement 1*).

We thus analysed suspected cholera cases, cholera stool cultures and case-area targeted interventions (CATIs) recorded between January 1, 2015, and December 31, 2017. Baseline characteristics of these data are summarized in *Appendix 1—table 1*.

**Appendix 1—table 1.** Baseline characteristics of suspected cholera cases, cholera stool cultures and case-area targeted interventions (CATIs) from January 1, 2015, to December 31, 2017.

| Suspected cholera cases | |
| --- | --- |
| Total no. of cases | 10931 |
| Median age (IQR) | 18 (35) |
| Sex ratio (M/F) | 1.0 |
| No. of cases with a comprehensive location (%) | 10428 (95%) |
| No. of different localities | 1497 |
| No. of cases with IV rehydration (%) | 2301 (21%) |
| No. of cases with a comprehensive location and IV rehydration (%) | 2144 (20%) |
| **Stool cultures** | |
| Total no. of stool samples cultured | 1070 |
| No. of stool cultures positive for *V. cholerae* O1 (%) | 509 (48%) |
| No. of positive cultures with a comprehensive location (%) | 360 (34%) |
| No. of different localities | 176 |
| **Case-area targeted interventions (CATIs)** | |
| Total no. of CATIs | 3887 |
| No. of CATIs conducted with EMIRA staff (%) | 2719 (70%) |
| No. of CATIs with a comprehensive location (%) | 3533 (91%) |
| No. of different localities | 815 |
| No. of CATIs with reported house decontamination (%) | 3655 (94%) |
| No. of decontaminated houses per CATI, median (IQR) | 4 (5) |
| No. of CATIs with reported education (%) | 3815 (98%) |
| No. of educated people per CATI, median (IQR) | 30 (47) |
| No. of CATIs with reported chlorine distribution (%) | 3748 (96%) |
| No. of household receiving chlorine per CATI, median (IQR) | 7 (8) |
| No. of CATIs with reported antibiotic prophylaxis (%) | 2002 (52%) |
| No. of people receiving antibiotic prophylaxis per CATI, median (IQR) | 20 (19) |
| No. of complete CATIs (%) | 3596 (93%) |
| No. of complete CATIs with antibiotic prophylaxis (%) | 1922 (49%) |

EMIRA, cholera rapid response team of the Ministry of health; IQR, interquartile range.
*Complete CATI, at least decontamination, education and distribution of chlorine tablets.

## Appendix 2

### Assessment of confounding by indication

In the event that case-area targeted interventions (CATIs) were significantly more likely implemented in more severe outbreaks, estimates of CATI effectiveness could be underestimated by a confounding by indication (*Remschmidt et al., 2015*).

Therefore, we initially compared baseline characteristics of outbreaks that were responded to ($\geq$1 complete CATI implemented within 7 days after the last recorded case of the outbreak) and outbreaks that were not. This outcome variable following a binomial distribution, univariate logistic mixed models were used to estimate odds ratios associated with each covariate. Localities were modelled as a random effect variable, in order to take into account the heterogeneity between localities.

We then evaluated CATI effectiveness ($CE_5$) by comparing the number of cases from the 4th day of outbreak between outbreaks that were and were not responded to. For this, we used logistic mixed models with cases from the 4th day of outbreak as a dependent variable (binomial distribution) (*Berridge and Crouchley, 2011*). We estimated the crude CATI effectiveness ($cCE_5$) as: 1 – Odds ratio. We then obtained adjusted estimates of CATI effectiveness ($aCE_5$) by adjusting for the following covariates: the number of cases and the number of positive cultures during the first 3 days of outbreak, rainfall, population density, travel time to the nearest town, accumulated case incidence between 2010 and 2014, coverage of OCV campaigns between 2012 and 2014 OCV campaigns, and the number of semesters since the beginning of the study.

A second evaluation of CATI effectiveness ($CE_6$) was performed by comparing the duration of outbreaks between outbreaks that were and were not responded to, using survival analyses censoring outbreak extinction. We fitted Cox models for Andersen-Gill counting process (AG-CP) (*Andersen and Gill, 1982*). We estimated the crude CATI effectiveness ($cCE_6$) as: 1 – (1/ Hazard ratio). We then obtained adjusted estimates of CATI effectiveness ($aCE_6$) by adjusting for confounders for which p-values were less than 0.25 at the initial univariate step (*Mickey and Greenland, 1989*).

For all effectiveness analyses, a p-value of less than 0.05 (two-sided) was considered to indicate statistical significance.

Overall, 238 identified outbreaks (53%) received a complete CATI within 7 days after the last recorded case, while 214 did not (*Figure 2*, *Appendix 2—table 1*). The proportion of outbreaks that were responded to progressively increased along the study semesters. These outbreaks occurred in localities which were significantly more densely populated, were significantly closer to a town, had significantly been less targeted by a previous mass OCV campaign than localities of outbreaks were non-responded to (*Appendix 2—table 1*). Outbreaks that were responded to exhibited a more severe onset (numbers of suspected cholera cases and positive cultures during the first 3 days were significantly higher), and they tended to be preceded by more cases than outbreaks that received no response (*Appendix 2—table 1*). This indicated a significant confounding by indication. Outbreaks that were responded to appeared to be more frequently preceded by complete CATIs than outbreaks that received no response (*Appendix 2—table 1*), which suggested a higher propensity of response.

**Appendix 2—table 1.** Baseline characteristics of outbreaks that were and were not responded to.

| | All outbreaks | Outbreaks with no complete CATI | Outbreaks with ≥ 1 complete CATI | Odds radio[†] (95% CI) | p-value[†] |
|---|---|---|---|---|---|
| No. of outbreaks | 452 | 214 (47%) | 238 (53%) | | |
| Semester since January 2015 | | | | 2.03 (1.63 to 2.51) | <0.0001* |
| Population density, median (IQR; inhab./km2) | 3.5 (6.5) | 3.4 (4.8) | 3.6 (11.5) | 1.01 (1 to 1.02) | 0.0308* |
| Travel time to the nearest town, median (IQR; minutes) | 26.7 (33.2) | 30.2 (31.9) | 24.9 (31.8) | 0.99 (0.98 to 1) | 0.0143* |
| Accumulated incidence between 2010 and 2014, median (IQR; per 1000inhabitants) | 103.8 (77.5) | 103.8 (293.4) | 103.8 (77.5) | 0.43 (0.08 to 2.33) | 0.327 |
| Coverage of OCV campaigns between 2012 and 2014, median (IQR; %) [mean, SD] | 0% (86) [25%, 40] | 0% (86) [30%, 42] | 0% (0) [21%, 38] | 0.52 (0.3 to 0.87) | 0.0137* |
| Previous cases in the same locality during the study, median (IQR; no. per year) | 4.3 (10.1) | 4 (9) | 5.2 (10.5) | 1.01 (0.99– 1.04) | 0.2320 |
| Previous complete CATIs in the same locality during the study, median (IQR; no. per year) | 0.2 (1.9) | 0 (0.8) | 0.9 (2.7) | 1.23 (1.11– 1.36) | <0.0001* |
| Daily rainfall during outbreak, median (IQR; mm) | 6.6 (13.3) | 5.2 (12.5) | 7.7 (13.3) | 1.01 (0.99 to 1.03) | 0.359 |
| No. of cases during the first 3 daysof outbreak, median (IQR) [mean, SD] | 2 (1) [2.5, 1.5] | 2 (0) [2.3, 0.9] | 2 (1) [2.7, 1.9] | 1.22 (1.04 to 1.43) | 0.0156* |
| No. of positive culture during the first 3 daysof outbreak, median (IQR) [mean, SD] | 0 (0) [0.2, 0.6] | 0 (0) [0.2, 0.5] | 0 (0) [0.3, 0.7] | 1.64 (1.12 to 2.39) | 0.0101* |

CATI, case-area targeted intervention; SD, standard deviation.

[†]Univariate comparisons using univariate logistic mixed models with response as model outcome and a binomial distribution, and outbreaks with no complete CATI as the reference class.

*Significant p-value.

Outbreaks that were responded to exhibited a paradoxically worse outcome than outbreaks that received no response. The median number of cases from the 4th day was 1 (interquartile range [IQR], 5) and 0 (IQR, 2) in outbreaks that were and were not responded to, respectively (*Appendix 2—figure 1A*, *Appendix 2—table 2*). Whereas the median duration of outbreaks that received at least one complete CATI was 11 days (IQR, 26.8), it was 3 days (IQR, 8) for outbreaks that were not responded to (*Appendix 2—figure 1B*, *Appendix 2—table 3*). The distribution of the number of cases from the 4th day of outbreaks that received no response (*Appendix 2—figure 1A*) looked like the distribution of the number of cases from the 4th day of outbreaks that were responded to within ≤ 1 days (*Figure 4A*), and looked like the distribution of the number of cases from the 4th day of outbreaks that received ≥ 1 CATI per week (*Figure 4C*). The Kaplan-Meier curve of outbreaks that received no response (*Appendix 2—figure 1B*) also looked like the Kaplan-Meier curve of outbreaks that were responded to within ≤ 1 days (*Figure 4B*), and looked like the Kaplan-Meier curve of outbreaks that received ≥ 1 CATI per case (*Figure 4D*). This illustrates the effect of the confounding by indication: because outbreaks that were not responded to were initially less severe, their outcome appeared better than the outcome of outbreaks that were responded to; the outcome of outbreaks that were not responded to also appeared close to the outcome of outbreaks that received a prompt and intense response.

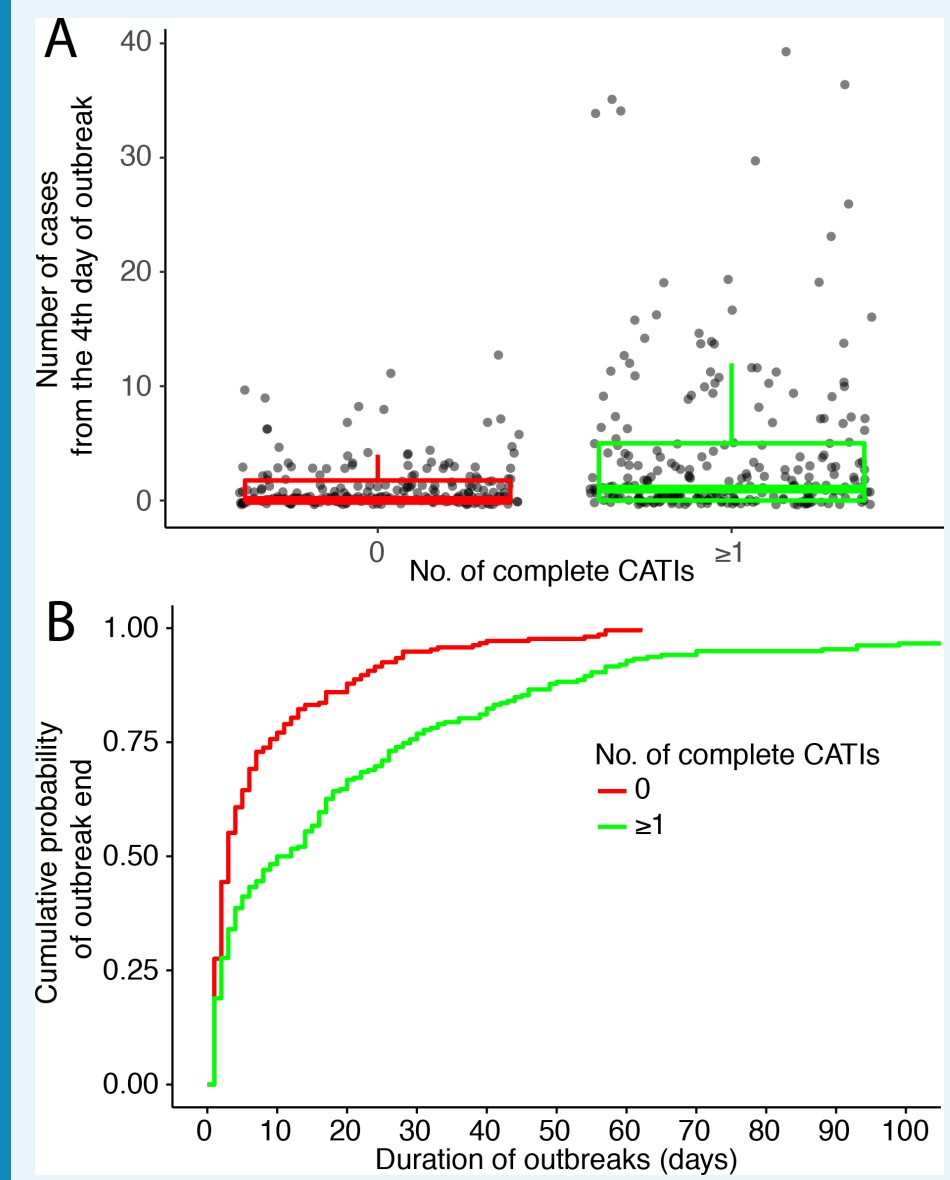

**Appendix 2—figure 1.** Outbreak outcome of outbreaks that were and were not responded to: (**A**) comparison of the outbreak size (number of suspected cholera cases from the 4th day of outbreak) and (**B**) Kaplan-Meier comparison of the outbreak duration (in days).

**Appendix 2—table 2.** Protective effectiveness of response: comparison of the number of suspected cholera cases from the 4th day of outbreak between outbreaks that were and were not responded to ($CE_5$).

| | | No. of cases from the 4th day of outbreak | | Crude estimate of CATI effectiveness ($cCE_5$)[†] | | Adjusted estimate of CATI effectiveness ($aCE_5$)[‡] | |
| | N | Median (IQR) | % (95% CI) | p-value | % (95% CI) | p-value |
|---|---|---|---|---|---|---|
| No. of CATIs during outbreak | | | | | | |
| No CATI[*] | 214 | 0 (2) | Ref | Ref | Ref | Ref |
| ≥1 CATIs | 238 | 1 (5) | −228% (−353 to −138) | <0.0001 | −411% (−638 to −254) | <0.0001 |

CATI, case-area targeted intervention; IQR, interquartile range.

[*]Reference class.

[†]Crude CATI effectiveness ($cCE_3$) was estimated on the No. of cases from the 4th day of outbreak, using logistic mixed models, as (1 – Odds ratio).

[‡]Estimates of CATI effectiveness ($aCE_3$) were adjusted according to covariates for which p-values were less than 0.25 at the initial univariate step (**Appendix 2—table 1**): number of cases and number of positive cultures during the first 3 days of outbreak, population density, travel time to the nearest town, coverage of OCV campaigns between 2012 and 2014 OCV campaigns, and semesters.

**Appendix 2—table 3.** Protective effectiveness of response: comparison of the duration of outbreaks between outbreaks that were and were not responded to ($CE_6$).

| | N | Duration of outbreak Median (IQR; days) | Crude estimate of response effectiveness ($cCE_6$)[†] % (95% CI) | p-value | Adjusted estimate of response effectiveness ($aCE_6$)[‡] % (95% CI) | p-value |
|---|---|---|---|---|---|---|
| No. of CATIs during outbreak | | | | | | |
| No CATI[*] | 214 | 3 (8) | Ref | Ref | Ref | Ref |
| ≥1 CATIs | 238 | 11 (26.75) | −319% (−457 to −216) | <0.0001 | −300% (−441 to −196) | <0.0001 |

CATI, case-area targeted intervention; IQR, interquartile range.

[*]Reference class.

[†]Crude response effectiveness ($CCE_4$) was estimated on the duration of outbreak, using Cox models for repeated events with Anderson-Gills correction (AGCP), as (1–1/hazard ratio).

[‡]Estimates of CATI effectiveness ($aCE_4$) were adjusted according to covariates for which p-values were less than 0.25 at the initial univariate step (**Appendix 2—table 1**): number of cases and number of positive cultures during the first 3 days of outbreak, population density, travel time to the nearest town, coverage of OCV campaigns between 2012 and 2014 OCV campaigns, and semesters.

Consequently, the crude CATI effectiveness in reducing the number of cases from the 4th day of outbreak ($cCE_5$) was estimated to be −228% (95% CI, −353 to −138), and after adjusting for potential confounders ($aCE_5$), −411% (−638 to −254) (**Appendix 2-table 2**). The crude CATI effectiveness on the duration of outbreak ($cCE_6$) was −319% (−457 to −216), and after adjusting for potential confounders ($aCE_6$), −300% (−441 to −196) (**Appendix 2—table 3**). This confirmed the significant confounding by indication, which explained why outbreaks that were responded to paradoxically exhibited a worse outcome than outbreaks that received no response (**Remschmidt et al., 2015**). This may be explained by the fact that numerous little outbreaks ended automatically, often before mobile teams arrived for the response. In absence of randomization, response teams likely tended to give priority to initially more severe outbreaks.

## Sensitivity analyses of CATI effectiveness

To assess the potential impact of our choices of definitions for outbreaks and for complete CATIs, we conducted several sensitivity analyses of CATI effectiveness according to different outbreak definitions, response definitions, and methods of model adjustment.

## Appendix 3.1

### Alternative outbreak definitions

In the main analyses, a cholera outbreak was defined by the occurrence, within the same locality, of at least two suspected cholera cases with at least one severely dehydrated case or positive culture, during a three-day time window, and after a refractory period of at least 21 days with no case. Outbreak onset was defined as the date of the first suspected case or positive culture, and outbreak end as the date of the last case or positive culture before a refractory period of at least 21 days. The three-day time window was initially chosen as it roughly corresponds to twice the median time incubation period of cholera (**Azman et al., 2013**). The 21 day refractory period was initially chosen as it roughly corresponds to twice the maximum incubation period (**Azman et al., 2013**) after the end of symptoms in the last case.

Nevertheless, using various thresholds of suspected cases, severely dehydrated cases, positive cultures, time window, and refractory period, we thus tested several alternative outbreak definitions summarized in **Appendix 3—table 1**.

**Appendix 3—table 1.** Alternative outbreak definitions.

| Cholera outbreak | Definition | Remark | No. of outbreaks |
|---|---|---|---|
| Outbreak A | • suspected cholera cases ≥ 2<br>• (severely dehydrated case + positive culture)≥1<br>• onset time window = 3 days<br>• refractory period = 21 days | Scenario 1<br>Main manuscript<br>Appendix 2<br>Appendix 3.2<br>Appendix 3.3 | 452 |
| Outbreak Cu0 | • suspected cholera cases ≥ 2<br>• irrespective of severely dehydrated cases and positive cultures<br>• onset time window = 3 days<br>• refractory period = 21 days | Scenario 2 | 2043 |
| Outbreak Cu1 | • suspected cholera cases ≥ 2<br>• positive culture ≥ 1<br>• irrespective of severely dehydrated cases<br>• onset time window = 3 days<br>• refractory period = 21 days | Scenario 3 | 64 |
| Outbreak Ca1 | • suspected cholera cases ≥ 1<br>• (severely dehydrated case + positive culture)≥1<br>• onset time window = 3 days<br>• refractory period = 21 days | Scenario 4 | 1514 |
| Outbreak T1 | • same as *Outbreak A* except:<br>• onset time window = 1 day | Scenario 5 | 267 |
| Outbreak T2 | • same as *Outbreak A* except:<br>• onset time window = 2 days | Scenario 6 | 394 |
| Outbreak T4 | • same as *Outbreak A* except:<br>• onset time window = 4 days | Scenario 7 | 494 |
| Outbreak T5 | • same as *Outbreak A* except:<br>• onset time window = 5 days | Scenario 8 | 535 |

*Appendix 3—table 1 continued on next page*

*Appendix 3—table 1 continued*

| Cholera outbreak | Definition | Remark | No. of outbreaks |
|---|---|---|---|
| Outbreak R7 | • same as *Outbreak A* except:<br>• refractory period = 7 days | Scenario 9 | 519 |
| Outbreak R14 | • same as *Outbreak A* except:<br>• refractory period = 14 days | Scenario 10 | 486 |

In order to estimate the confounding by indication, we then compared outbreak outcome between outbreaks that were and were not responded to, as described in Appendix 2. In order to estimate CATI effectiveness, we compared outbreak outcome between classes of response promptness and between classes of response intensity, as described in the main manuscript.

Overall, all outbreak definitions led to a significant confounding by indication (***Appendix 3—table 2***). When considering only outbreaks that were responded to, CATI effectiveness according to response promptness and response intensity on the reduction of accumulated cases and on the reduction of outbreak duration remained consistent, irrespective of the adopted outbreak definition (***Appendix 3—table 2***). Some alternative outbreak definitions even brought higher effectiveness estimates than the definition used in the main manuscript.

**Appendix 3—table 2.** Sensitivity analysis on outbreak and CATI definitions.

| Scenario | 1 | 2 | 3 | 4 | 5 | 6 | 7 | 8 | 9 | 10 |
|---|---|---|---|---|---|---|---|---|---|---|
| Outbreak definition[*] | Outbreak A | Outbreak Cu0 | Outbreak Cu1 | Outbreak Ca1 | Outbreak T1 | Outbreak T2 | Outbreak T4 | Outbreak T5 | Outbreak R7 | Outbreak R14 |
| Response definition[†] | CATIc7 | CATIc7 | CATIc7 | CATIc7 | CATIc7 | CATIc7 | CATIc7 | CATIc7 | CATIc7 | CATIc7 |
| No. of outbreaks | 452 | 2043 | 64 | 1514 | 267 | 394 | 494 | 535 | 519 | 486 |
| No. of CATIs | 3596 | 3596 | 3596 | 3596 | 3596 | 3596 | 3596 | 3596 | 3596 | 3596 |
| No. of CATIs during outbreaks (%) | 633 (18%) | 1445 (40%) | 153 (4%) | 1000 (28%) | 386 (11%) | 540 (15%) | 670 (19%) | 717 (20%) | 497 (14%) | 576 (16%) |
| No. of outbreaks that were responded to (%) | 238 (53%) | 730 (36%) | 45 (70%) | 500 (33%) | 152 (57%) | 211 (54%) | 256 (52%) | 276 (52%) | 242 (47%) | 240 (49%) |
| Comparison between outbreaks that were and were not responded to | | | | | | | | | | |
| No. of cases during the first 3 days of outbreak, Odds ratio (95% CI) | 1.22 (1.04 to 1.43) | 1.61 (1.43 to 1.8) | 24.69 (1.4 to 435.42) | 1.69 (1.47 to 1.96) | 1.1 (0.85 to 1.43) | 1.15 (0.96 to 1.37) | 1.19 (1.04 to 1.36) | 1.19 (1.05 to 1.34) | 1,25 (1,07 to 1,45) | 1,24 (1,06 to 1,45) |
| No. of positive culture during the first 3 days of outbreak, Odds ratio (95% CI) | 1.64 (1.12 to 2.39) | 2.29 (1.63 to 3.21) | 1.82 (0.64 to 5.21) | 2.16 (1.61 to 2.9) | 1.35 (0.77 to 2.36) | 1.6 (1.06 to 2.41) | 1.85 (1.26 to 2.72) | 1.85 (1.29 to 2.65) | 2,08 (1,42 to 3,06) | 1,91 (1,3 to 2,8) |

*Appendix 3—table 2 continued on next page*

*Appendix 3—table 2 continued*

| Scenario | 1 | 2 | 3 | 4 | 5 | 6 | 7 | 8 | 9 | 10 |
|---|---|---|---|---|---|---|---|---|---|---|
| CATI effectiveness according to the response promptness | | | | | | | | | | |
| ≤1 day vs >7 days crude estimate of CATI effectiveness on accumulated cases (95% CI) ($cCE_1$) | 83% (71 to 90) | 85% (79 to 89) | 40% (−86 to 81) | 89% (83 to 92) | 73% (48 to 87) | 79% (63 to 88) | 86% (86 to 87) | 84% (70 to 91) | 89% (69 to 96) | 86% (72 to 93) |
| ≤1 day vs >7 days crude estimate of CATI effectiveness on outbreak duration (95% CI) ($cCE_2$) | 59% (36 to 74) | 74% (66 to 80) | 22% (−81 to 66) | 78% (69 to 84) | 48% (18 to 67) | 54% (29 to 70) | 61% (40 to 75) | 58% (34 to 73) | 65% (39 to 80) | 62% (43 to 75) |
| CATI effectiveness according to the response intensity | | | | | | | | | | |
| ≥1 vs <0.25 CATIs per week crude estimate of CATI effectiveness on accumulated cases (95% CI) ($cCE_3$) | 74% (44 to 88) | 77% (62 to 85) | 74% (−122 to 97) | 69% (43 to 84) | 28% (−85 to 72) | 69% (33 to 86) | 82% (60 to 92) | 83% (61 to 92) | 87% (86 to 87) | 66% (−10 to 89) |
| ≥1 vs <0.25 CATIs per case crude estimate of CATI effectiveness on outbreak duration (95% CI) ($cCE_4$) | 76% (54 to 88) | 89% (85 to 92) | 55% (−155 to 92) | 91% (86 to 94) | 89% (79 to 95) | 81% (66 to 89) | 76% (55 to 87) | 75% (54 to 86) | 78% (59 to 88) | 81% (68 to 88) |

[*]see *Appendix 3—table 1*.

[†]see *Appendix 3—table 2*.

# Appendix 3.2

## Alternative response definitions

In the main analyses, cholera outbreaks were considered as responded to if they received at least one complete CATI (i.e. a case-area targeted intervention with at least education, house decontamination by spraying and distribution of chlorine tablets) within 7 days after the last recorded case of the outbreak.

We thus tested several alternative response definitions summarized in *Appendix 3—table 3*.

**Appendix 3—table 3.** Alternative response definitions.

| Cholera outbreak | Definition | Remark | No. of CATIs | No. of CATIs during outbreaks (%) |
|---|---|---|---|---|
| CATIc7 | • Complete CATI (mobile teams reported at least education, house decontamination by spraying and distribution of chlorine tablets) <br> • Implemented within 7 days after the last recorded case of the outbreak | Scenario 1 <br> Main manuscript <br> Appendix 2 <br> Appendix 3.1 <br> Appendix 3.3 | 3596 | 633 (18%) |
| CATIc0 | • Complete CATI <br> • implemented before the last recorded case of the outbreak | Scenario 11 | 3596 | 501 (14%) |
| CATI7 | • All CATI (irrespective of activities reported by mobile teams <br> • Implemented within 7 days after the last recorded case of the outbreak | Scenario 12 | 3887 | 681 (18%) |
| CATIcEMIRA7 | • Complete CATI <br> • Conducted by NGO mobile teams in tandem with EMIRA staff <br> • Implemented within 7 days after the last recorded case of the outbreak | Scenario 13 | 2539 | 458 (18%) |
| CATIcATB7 | • Complete CATI and reported antibiotic prophylaxis <br> • Implemented within 7 days after the last recorded case of the outbreak | Scenario 14 <br> Main manuscript | 1922 | 350 (18%) |

EMIRA, cholera rapid response team of the Ministry of health.

In order to estimate the confounding by indication, we then compared outbreak outcome between outbreaks that were and were not responded to, as described in Appendix 2. In order to estimate CATI effectiveness, we compared outbreak outcome between classes of response promptness and between classes of response intensity, as described in the main manuscript.

Overall, all response definitions led to a significant confounding by indication (*Appendix 3—table 4*). When considering only outbreaks that were responded to, CATI effectiveness according to response promptness and response intensity on the reduction of accumulated cases and on the reduction of outbreak duration remained consistent, irrespective of the adopted response definition (*Appendix 3—table 4*). Some alternative response definitions even brought higher effectiveness estimates than the definition used in the main manuscript.

**Appendix 3—table 4.** Sensitivity analysis on outbreak and CATI definitions.

| Scenario | 1 | 11 | 12 | 13 | 14 |
|---|---|---|---|---|---|
| Outbreak definition[*] | Outbreak A | Outbreak A | Outbreak A | Outbreak A | Outbreak A |

*Appendix 3—table 4 continued on next page*

*Appendix 3—table 4 continued*

| Scenario | 1 | 11 | 12 | 13 | 14 |
|---|---|---|---|---|---|
| Response definition[†] | CATIc7 | CATIc0 | CATI7 | CATIcEMIRA7 | CATIcATB7 |
| No. of outbreaks | 452 | 452 | 452 | 452 | 452 |
| No. of CATIs | 3596 | 3596 | 3887 | 2539 | 1922 |
| No. of CATIs during outbreaks (%) | 633 (18%) | 501 (14%) | 681 (18%) | 458 (18%) | 350 (18%) |
| No. of outbreaks that were responded to (%) | 238 (53%) | 172 (38%) | 244 (54%) | 201 (44%) | 160 (35%) |
| Comparison between outbreaks that were and were not responded to | | | | | |
| No. of cases during the first 3 days of outbreak, Odds ratio (95% CI) | 1.22 (1.04 to 1.43) | 1.49 (1.24 to 1.80) | 1.22 (1.04 to 1.43) | 1.21 (1.04 to 1.40) | 1.09 (0.97 to 1.24) |
| No. of positive culture during the first 3 days of outbreak, Odds ratio (95% CI) | 1.64 (1.12 to 2.39) | 1.63 (1.14 to 2.32) | 1.62 (1.11 to 2.37) | 2.12 (1.44 to 3.11) | 1.92 (1.35 to 2.73) |
| CATI effectiveness according to the response promptness | | | | | |
| ≤1 day vs >7 days crude estimate of CATI effectiveness on accumulated cases (95% CI) ($cCE_1$) | 83% (71 to 90) | 85% (76 to 91) | 83% (71 to 90) | 89% (81 to 94) | 90% (81 to 94) |
| ≤1 day vs >7 days crude estimate of CATI effectiveness on outbreak duration (95% CI) ($cCE_2$) | 59% (36 to 74) | 65% (44 to 78) | 57% (33 to 72) | 76% (61 to 85) | 84% (74 to 90) |
| CATI effectiveness according to the response intensity | | | | | |
| ≥1 vs <0.25 CATIs per week crude estimate of CATI effectiveness on accumulated cases (95% CI) ($cCE_3$) | 74% (44 to 88) | 81% (64 to 90) | 70% (35 to 86) | 77% (49 to 90) | 85% (64 to 94) |
| ≥1 vs <0.25 CATIs per case crude estimate of CATI effectiveness on outbreak duration (95% CI) ($cCE_4$) | 76% (54 to 88) | 54% (−14 to 82) | 75% (53 to 86) | 86% (72 to 93) | 86% (71 to 93) |

[*]see ***Appendix 3—table 1***.

[†]see ***Appendix 3—table 3***.

# Appendix 3.3

## Alternative adjustment methods for effectiveness estimates

In the main analyses, we adjusted CATI effectiveness estimates for covariates for which p-values were less than 0.25 at the initial univariate step (*Mickey and Greenland, 1989*).

We thus tested two alternative methods of confounder selection: adjusting on all eight covariates (number of cases and number of positive cultures during the first 3 days of outbreak, rainfall, population density, travel time to the nearest town, accumulated case incidence between 2010 and 2014, coverage of OCV campaigns between 2012 and 2014 OCV campaigns, and number of semesters since the beginning of the study); and minimizing the Akaike information criterion (AIC) of the models in order to avoid overfitting (***Appendix 3—table 5***).

**Appendix 3—table 5.** Alternative adjustment methods for CATI effectiveness estimates.

| | Crude estimates | | Estimates adjusted for covariates selected by p-values† | | Estimates adjusted for all covariates | | Estimates adjusted for covariates selected by AIC* | |
|---|---|---|---|---|---|---|---|---|
| | No. of covariates (AIC*) | cCE* | No. of covariates (AIC*) | aCE* | No. of covariates (AIC*) | aCE* | No. of covariates (AIC*) | aCE* |
| CATI effectiveness according to the response promptness | | | | | | | | |
| ≤1 day vs >7 days estimate of CATI effectiveness on accumulated cases (95% CI) (CE$_1$) | 0 (1102.35) | 83% (71 to 90) | 5 (1 096.91) | 76% (59 to 86) | 8 (1073.75) | 77% (62 to 87) | 6 (1072.51) | 79% (65 to 88) |
| ≤1 day vs >7 days crude estimate of CATI effectiveness on outbreak duration (95% CI) (CE$_2$) | 0 (956.25) | 59% (36 to 74) | 5 (933.42) | 61% (41 to 75) | 8 (929.93) | 65% (46 to 77) | 5 (924.20) | 65% (46 to 77) |
| CATI effectiveness according to the response intensity | | | | | | | | |
| ≥1 vs <0.25 CATIs per week estimate of CATI effectiveness on accumulated cases (95% CI) (CE$_3$) | 0 (1123.91) | 74% (44 to 88) | 2 (1112.44) | 58% (8 to 81) | 8 (1093.38) | 62% (18 to 82) | 7 (1092.17) | 62% (22 to 81) |
| ≥1 vs <0.25 CATIs per case estimate of CATI effectiveness on outbreak duration (95% CI) (CE$_4$) | 0 (945.29) | 76% (54 to 88) | 3 (949.59) | 73% (49 to 86) | 8 (931.29) | 76% (56 to 87) | 3 (925.52) | 54% (27 to 71) |

*AIC, Akaike information criterion; CATI, case-area targeted intervention; cCE, crude CATI effectiveness estimates; aCE, adjusted CATI effectiveness estimates.

†Covariates for which p-values were less than 0.25 at the initial univariate step (**Tables 1** and **2**).

Overall, all adjustment methods of models led to consistent CATI effectiveness estimates (**Appendix 3—table 5**). Inclusion of all covariates did not bring important overfitting.

