## [Decision Letter]

**Acceptance summary:**

This manuscript addresses an important and timely question around the effectiveness of case-area targeted interventions (CATIs) against cholera in Haiti. The cholera epidemic in Haiti has been swift and devastating. It has led to substantial scientific study of the spatio-temporal dynamics of the disease, but also to fundamental debates on transmission routes and implications for best approaches to control and possibly elimination. The detailed monitoring of CATIs in the Centre Department provides an opportunity for real, on-the-ground, assessment of this strategy. With a set of robust statistical analyses, the manuscript presents evidence for the effectiveness and importance of CATIs. It finds that a prompter response (measured in categories of days since the first case was reported) and a more intense response (measured in the number of CATIs per week and/or case of the outbreak) were effective at reducing the number of cases and duration of cholera outbreaks.

**Decision letter after peer review:**

Thank you for submitting your article "Estimating effectiveness of case-area targeted response interventions against cholera in Haiti" for consideration by *eLife*. Your article has been reviewed by three peer reviewers, and the evaluation has been overseen by a Reviewing Editor and Neil Ferguson as the Senior Editor. The following individuals involved in review of your submission have agreed to reveal their identity: Virginia E. Pitzer.

The reviewers have discussed the reviews with one another and the Reviewing Editor has drafted this decision to help you prepare a revised submission.

Overall, we feel that the results are novel and important for justifying the continued (and expanded) use of CATIs in response to cholera. The statistical analysis was also felt to be through and well handled. However, we have concerns about the methodological approach, interpretation of findings and presentation of results.

Essential revisions:

1) Our major concern was the choice of CATIs to include. For instance, according to the manuscript, only 15% of the CATI interventions were done in response to an "outbreak" detected (a-posteriori). Thus it seems like CATI that weren't implemented within an outbreak were excluded from the analysis but it was maybe just the effect of those CATI that kept the number of cases small enough to stay below the authors definition of an "outbreak". This potentially adds important bias to the reported results. While this limitation might be inherent in a retrospective study, the manuscript needs to acknowledge this limitation more clearly. In particular, you should discuss how these excluded CATIs may affect the results.

2) If the data allow to differentiate between different types of CATI (for instance antibiotics) then an additional stratified analysis of CATIs with or without prophylactic antibiotics should be included. For instance, the findings of the study would be considerably strengthened by including an additional analysis of the association between the promptness and intensity of the response and the outbreak size and duration stratified by whether or not the CATIs included the distribution of prophylactic antibiotics to household members of identified cases. Based on the results presented in Appendix 3—table 4, it would seem that such an analysis would be feasible and straightforward to perform, and would have reasonable power to detect a potential difference in the effect estimates. Given the use of antibiotic prophylaxis to prevent cholera is somewhat controversial compared to the other components of the CATIs, it would be very interesting and potential impactful to have an estimate of its effectiveness from a real-world implementation setting.

3) More details on the CATIs would also be useful, for example: Do you know how many households or people have been targeted with exactly what intervention? Or in what geographical area around cases the interventions have been applied? The same applies to case data (you mention that some confirmed cases are not mentioned in the line list, and that many lab results are negative for *V. cholerae*).

4) Regarding the analysis of confounders and subsequent control for confounding in the adjusted models of intervention effectiveness: it's not clear why you went to all the trouble to analyse which covariates may be potential confounders of the relationship between response promptness (Table 1) and response intensity (Table 2) and outcomes of interest (outbreak size and duration), but then subsequently included all of the potential covariates in the adjusted models. By definition, any covariate that is not associated with both the exposure and outcome of interest is not a potential confounder. Furthermore, including covariates that are downstream of other confounders has the potential to inadvertently introduce bias into an analysis (see Hernan et al., Epidemiology, 2004). The sensitivity analysis presented in Appendix 3—table 5 shows that the results are very similar when only adjusting for the covariates that are significantly associated with the exposure and/or outcome. Nevertheless, these should be presented in the main text, and the model including all potential covariates could be included as a sensitivity analysis.

5) The presentation of results in the main text would be clearer and more concise if you focused the analysis of the association between the intensity of the response and the outbreak size and duration on only one of the exposure-outcome combinations. As is, you present the results for the association between response intensity as measured by the number of CATIs per week and the outcome as measured by the cumulative number of cholera cases in the outbreak in Table 5, and the association between response intensity as measured by the number of CATIs per case and the outcome as measured by the outbreak duration in Table 6. Why were these exposure-outcome combinations chosen and not the other two alternatives (e.g. the association between the number of CATIs per week and outbreak duration, and the association between the number of CATIs per case and outbreak size)? In general, the results were quite similar (although there was little difference between the effectiveness of having 0.25 to <0.5 CATIs per case and the baseline comparator of >0 to <0.25 CATIs per case). Therefore, just one of the results should be presented in the main text, and the other 3 exposure-outcome combinations presented in the Appendix, with only the qualitative differences described in the main text. Additionally, the notation used in the tables and figures for these categories is quite confusing and unnecessary.

6) Another concern is in regard to the interpretation of the potential impact of the under-diagnosis of cases and misclassification of non-cholera diarrhoea cases in paragraph four of the Discussion. Here, it is stated that both a high number of unrecorded cases and the misclassification of outbreaks due to inconsistent culture confirmation would lead to an underestimation of the effectiveness of CATIs. However, we do not believe there is any reason to conclude that the direction of the bias would be to underestimate the effect of the intervention in either instance. In order for the first statement to be true, cases would have to be differentially underdiagnosed more often when the CATIs are implemented earlier compared to when they were introduced >7 days after the outbreak started, or else cases would have to be underdiagnosed more often when the control efforts were more intense. Neither situation seems likely. While misclassification of non-cholera diarrhoea outbreaks could potentially attenuate the effectiveness estimate for the interventions (as suggested by the alternative outbreak definition "Outbreak Cu1" examined in Appendix 3—table 2), this was not true for most of the other outbreak definitions, and again is unlikely to be true more generally. At the very least, more justification should be provided for these statements.

7) The effectiveness of CATIs has implications well beyond the specific control question at hand, on the still poorly agreed importance of routes of transmission in cholera. In turn, these routes and in particular the role of an environmental reservoir continue to be the subject of substantial debate, and also influence views on approaches to control and sustain the gains of control in the long run. We understand that the paper is not meant to resolve these matters but it would be valuable to add some discussion of related implications even if briefly. These will broaden the scientific value of the findings.

8) There is concern about the choice of "outbreaks", which the authors define post-hoc based on reported incidence, as the unit of analysis. Some discussion would be useful of the implications if the unit of analysis is either an individual or a household, targeted with interventions or not.

9) Outbreaks that haven't been responded to by a targeted intervention have been excluded from the analysis because the authors have detected major bias ("confounding by indication"), e.g. larger outbreaks are more likely to be targeted by > 1 intervention. Doesn't the same bias exist between outbreaks that have been responded to more or less promptly or with a higher or lower intensity?

---

## [Author Response]

Essential revisions:1) Our major concern was the choice of CATIs to include. For instance, according to the manuscript, only 15% of the CATI interventions were done in response to an "outbreak" detected (a-posteriori). Thus it seems like CATI that weren't implemented within an outbreak were excluded from the analysis but it was maybe just the effect of those CATI that kept the number of cases small enough to stay below the authors definition of an "outbreak". This potentially adds important bias to the reported results. While this limitation might be inherent in a retrospective study, the manuscript needs to acknowledge this limitation more clearly. In particular, you should discuss how these excluded CATIs may affect the results.

We thank the reviewers for raising the concern of CATIs conducted outside identified outbreaks.

To clarify this point, we included the number and proportion of CATIs during outbreaks in Appendix 3—table 1, in Appendix 3—table 3 and in Appendix 3—table 4.

CATIs conducted outside identified outbreaks were implemented in response to sporadic cases that did not meet outbreak definition criteria, as illustrated by the higher proportions of CATIs within outbreaks found when using less stringent outbreak definitions (up to 40% in scenario 2, Appendix 3—table 2).

However, effectiveness sensitivity analyses using alternative outbreak definitions that included higher proportions of CATIs, did not exhibit substantially different results (Appendix 3—table 2).

Sporadic CATIs may theoretically have prevented, delayed or attenuated the emergence of certain outbreaks. They may also be associated with the propensity of future outbreak response. In our revised effectiveness analyses, we thus included the frequency of previous complete CATIs conducted between the beginning of the study period and the outbreak onset as a new covariate (see Table 1 and 2). Because previous CATIs may be dependent of the previous number of cases to be responded to, we also included the previous incidence of suspected cholera cases recorded between the beginning of the study period and the outbreak onset as another new covariate. However, we found no significant association between these covariates and response promptness or intensity. We now address this concern in the revised Discussion section.

Finally, sporadic CATIs may prevent the occurrence of future outbreaks, or attenuate outbreaks that may thus end before being responded. Therefore, the real impact of CATIs against cholera may in fact be broader than suggested by our analyses, which specifically assessed the effectiveness of prompt and intense CATIs to reduce the size and duration of outbreaks that received a response.

2) If the data allow to differentiate between different types of CATI (for instance antibiotics) then an additional stratified analysis of CATIs with or without prophylactic antibiotics should be included. For instance, the findings of the study would be considerably strengthened by including an additional analysis of the association between the promptness and intensity of the response and the outbreak size and duration stratified by whether or not the CATIs included the distribution of prophylactic antibiotics to household members of identified cases. Based on the results presented in Appendix 3—table 4, it would seem that such an analysis would be feasible and straightforward to perform, and would have reasonable power to detect a potential difference in the effect estimates. Given the use of antibiotic prophylaxis to prevent cholera is somewhat controversial compared to the other components of the CATIs, it would be very interesting and potential impactful to have an estimate of its effectiveness from a real-world implementation setting.

Our study initially did not aim at evaluating the respective effectiveness of each activity included in CATIs. Besides, stratified analyses on house decontamination, education and chlorine distribution was not possible as nearly all CATIs included these activities.

However, we thank reviewers for suggesting us to include stratified analyses on antibiotic prophylaxis, which are now described in a separate part in the Results section. We conducted additional comparisons of outbreak size or outbreak duration according to the response promptness or to the response intensity, stratified by whether all complete CATIs or none of complete CATIs conducted within outbreaks included antibiotic prophylaxis. We found that most of effectiveness estimates appeared higher when all CATIs included antibiotic prophylaxis.

We also discuss these results, including the risk of bacterial resistance selection when using antibiotics as selected chemoprophylaxis.

Nevertheless, we would like to point out that the methodology used in this study is probably not the most appropriate to accurately measure the impact of an antibiotic prophylaxis delivered during CATIs. Other approaches, such as randomization of antibiotic distribution during CATIs, would help to better understand the additional impact of this measure. The circumstances in which we conducted our study did not allow such randomization. This is why we insisted at the end of the Discussion on the interest of conducting additional research to better define the optimal modalities for the implementation of CATIs.

3) More details on the CATIs would also be useful, for example: Do you know how many households or people have been targeted with exactly what intervention? Or in what geographical area around cases the interventions have been applied? The same applies to case data (you mention that some confirmed cases are not mentioned in the line list, and that many lab results are negative for *V. cholerae*).

Following the reviewers’ request, we added a supplementary Table describing baseline characteristics of suspected cholera cases, cholera stool cultures and CATIs (Appendix 1—table 1), including the number of households targeted by decontamination or chlorine distribution and the number of people who were educated and who received antibiotic chemoprophylaxis.

4) Regarding the analysis of confounders and subsequent control for confounding in the adjusted models of intervention effectiveness: it's not clear why you went to all the trouble to analyse which covariates may be potential confounders of the relationship between response promptness (Table 1) and response intensity (Table 2) and outcomes of interest (outbreak size and duration), but then subsequently included all of the potential covariates in the adjusted models. By definition, any covariate that is not associated with both the exposure and outcome of interest is not a potential confounder. Furthermore, including covariates that are downstream of other confounders has the potential to inadvertently introduce bias into an analysis (see Hernan et al., Epidemiology, 2004). The sensitivity analysis presented in Appendix 3—table 5 shows that the results are very similar when only adjusting for the covariates that are significantly associated with the exposure and/or outcome. Nevertheless, these should be presented in the main text, and the model including all potential covariates could be included as a sensitivity analysis.

We agree with the reviewers’ suggestion and therefore reintegrated adjusted estimates for covariates for which P-values were less than 0.25 at the initial univariate step in the main manuscript. We now only present the model including all potential covariates as a sensitivity analysis (Appendix 3—table 5). We also kept models minimizing the Akaike information criterion (AIC) as a sensitivity analysis (Appendix 3—table 5). Results between the three model groups are indeed very similar.

We also totally agree that including covariates that are downstream of other confounders has the potential to inadvertently introduce bias into an analysis. We believe that the four final models presented in the main text do not present such bias.

5) The presentation of results in the main text would be clearer and more concise if you focused the analysis of the association between the intensity of the response and the outbreak size and duration on only one of the exposure-outcome combinations. As is, you present the results for the association between response intensity as measured by the number of CATIs per week and the outcome as measured by the cumulative number of cholera cases in the outbreak in Table 5, and the association between response intensity as measured by the number of CATIs per case and the outcome as measured by the outbreak duration in Table 6. Why were these exposure-outcome combinations chosen and not the other two alternatives (e.g. the association between the number of CATIs per week and outbreak duration, and the association between the number of CATIs per case and outbreak size)? In general, the results were quite similar (although there was little difference between the effectiveness of having 0.25 to <0.5 CATIs per case and the baseline comparator of >0 to <0.25 CATIs per case). Therefore, just one of the results should be presented in the main text, and the other 3 exposure-outcome combinations presented in the Appendix, with only the qualitative differences described in the main text. Additionally, the notation used in the tables and figures for these categories is quite confusing and unnecessary.

We thank the reviewers for these comments and better explained in the Materials and method section why we used two different proxies of response intensity when comparing outbreak size or outbreak duration. Indeed, we had to avoid that cases or duration be included both within outcome and exposure variables: we could neither compare the number of *cases* accumulated from the 4th day of outbreak between classes of CATIs per *case*, nor compare the *duration* of outbreaks between classes of CATIs per *week*. We thus approximated response intensity by the number of complete CATIs per week ratio when comparing the number of cases accumulated from the 4th day of outbreak (CE_3_), and by the number of complete CATIs per case ratio when comparing the duration of outbreak (CE_4_).

Considering that we independently evaluated CATI effectiveness by comparing outbreak outcome defined by two different variables (outbreak size and outbreak duration), between classes of two exposure variables (response promptness and response intensity), we believed important to keep all four groups of results within the main manuscript. The fact that all effectiveness estimates were of the same order of magnitude suggests that our results are consistent.

Finally, we clarified the notation for classes of response intensity used in the tables and figures, as suggested by the reviewers.

6) Another concern is in regard to the interpretation of the potential impact of the under-diagnosis of cases and misclassification of non-cholera diarrhoea cases in paragraph four of the Discussion. Here, it is stated that both a high number of unrecorded cases and the misclassification of outbreaks due to inconsistent culture confirmation would lead to an underestimation of the effectiveness of CATIs. However, we do not believe there is any reason to conclude that the direction of the bias would be to underestimate the effect of the intervention in either instance. In order for the first statement to be true, cases would have to be differentially underdiagnosed more often when the CATIs are implemented earlier compared to when they were introduced >7 days after the outbreak started, or else cases would have to be underdiagnosed more often when the control efforts were more intense. Neither situation seems likely. While misclassification of non-cholera diarrhoea outbreaks could potentially attenuate the effectiveness estimate for the interventions (as suggested by the alternative outbreak definition "Outbreak Cu1" examined in Appendix 3—table 2), this was not true for most of the other outbreak definitions, and again is unlikely to be true more generally. At the very least, more justification should be provided for these statements.

We thank the reviewers for this comment and acknowledge that our CATI effectiveness may have been over or under estimated depending on the differential distribution of missing values between classes of response promptness and intensity. We modified our Discussion accordingly.

Despite missing data, we believe that our sensitivity analyses using very different outbreak definitions provide sufficient confidence that prompt and intense CATIs have the potential to effectively reduce cholera outbreak size and duration.

7) The effectiveness of CATIs has implications well beyond the specific control question at hand, on the still poorly agreed importance of routes of transmission in cholera. In turn, these routes and in particular the role of an environmental reservoir continue to be the subject of substantial debate, and also influence views on approaches to control and sustain the gains of control in the long run. We understand that the paper is not meant to resolve these matters but it would be valuable to add some discussion of related implications even if briefly. These will broaden the scientific value of the findings.

We agree with the reviewers that better understanding the respective importance of cholera transmission routes and the potential role of an environment reservoir can impact strategic choices in epidemiological surveillance and control. But as we believe that household water treatment, sanitation and hygiene promotion, as well as antibiotic prophylaxis may theoretically prevent both human-to-human and environment-to-human cholera transmission pathways, we initially did not discuss this important issue within the specific context of this paper dedicated to case-area-targeted interventions. Nevertheless, following the reviewers’ suggestion, we now discuss this important scientific issue in our conclusion.

8) There is concern about the choice of "outbreaks", which the authors define post-hoc based on reported incidence, as the unit of analysis. Some discussion would be useful of the implications if the unit of analysis is either an individual or a household, targeted with interventions or not.

Together with the reviewers, we are concerned that our effectiveness estimates may be biased by the fact that our retrospectively defined outbreaks may be an approximate unit of analysis in term of space, time and population. This is why we conducted a sensitivity analysis using alternative definitions, including systematically lab-confirmed cholera outbreaks, which showed consistent and robust estimates (Appendix 3.1). As suggested by the reviewers, another CATI effectiveness studies at the household level is underway in Haiti.

We believe that our Discussion section now better acknowledges this potential bias.

9) Outbreaks that haven't been responded to by a targeted intervention have been excluded from the analysis because the authors have detected major bias ("confounding by indication"), e.g. larger outbreaks are more likely to be targeted by > 1 intervention. Doesn't the same bias exist between outbreaks that have been responded to more or less promptly or with a higher or lower intensity?

We shared the same concern as the reviewers regarding the possibility of a residual confounding by indication in the group of outbreaks that received a CATI response.

However, proxies of outbreak initial severity (no. of cases and no. of positive cultures during the first three days) were not systematically higher in classes of better response promptness or intensity. In addition, our models were adjusted for potential confounders and took into account the heterogeneity between localities. This quasi-experimental study was also stratified on response promptness and on response intensity, which classes yielded consistent response effectiveness estimates.

Anyway, in the event a significant but hidden confounding by indication remained, this should have underestimated CATI effectiveness, which all appeared significantly positive.